# Pan-cancer analysis and validation show GTF2E2's diagnostic, prognostic, and immunological roles in regulating ferroptosis in endometrial cancer

Nie Zhang[1,2,3], Xuejin Qin[1], Jingjing Liu[1], Ke Han[1], Manman Kang[1], Zhengchun Zhu[1], Di Zhang[1], Fei Zhong [1,2,3]*

1 Department of Oncology, Fuyang Hospital of Anhui Medical University, Fuyang, China, 2 Graduate School of Anhui Medical University, Anhui, China, 3 Key Laboratory of Gametes and Abnormal Reproductive Tract of National Health Commission, Anhui Medical University, Anhui, China

* zhongfei@ahmu.edu.cn (FZ)

## Abstract

### Background

Transcription initiation factor IIE subunit beta (GTF2E2) is a crucial component of the RNA polymerase II transcription initiation complex. There is a lack of more detailed research on the biological function of GTF2E2 in pan-cancer.

### Methods

We conducted a comprehensive pan-cancer analysis using data from The Cancer Genome Atlas (TCGA) and Genotype-Tissue Expression (GTEx) project. Employing a multi-pronged approach with tools including R, Cytoscape, TISIDB, cBioPortal, STRING, GSCALite, and CancerSEA, we investigated GTF2E2's expression patterns, prognostic value, mutational landscape, functional enrichment, and immunological associations across 33 cancer types. Besides, we further validated the bioinformatic results through in vitro experiments in Uterine corpus endometrial carcinoma (UCEC), including western blotting (WB), cell proliferation assays and transwell. DCFH-DA, C11-BODIPY 581/591 and FeRhoNox-1 probes were performed to identify ferroptosis levels in vitro.

### Results

GTF2E2 expression was significantly elevated in most cancers compared to normal tissues, with notable diagnostic potential (AUC > 0.7) in 20 cancer types. GTF2E2 expression varied across molecular and immune subtypes and correlated with tumor stage and patient age in several cancers. Functional enrichment analyses highlighted GTF2E2's involvement in key cancer-related and immunological pathways. Notably, GTF2E2 promoted UCEC progression in vitro, and knockdown of GTF2E2 significantly inhibited the proliferation, migration and invasion of UCEC cells. Compared with the control group, GPX4 expression was down-regulated and ACSL4 expression was up-regulated in the

**Data availability statement:** All relevant data are within the paper and its Supporting information files.

**Funding:** This work was supported by the Anhui University Natural Science Research Project (2023AH050675); Research Fund of Anhui Institute of Translational Medicine (2023zhyx-C98); Research Fund Project of Anhui Medical University (2021xkj077, 2022xkj213); Anhui Provincial Key Research and Development Project (2022e07020050); Science Research Project of Anhui Health Commission (AHWJ2021b097); Science Research Project of Anhui Health Commission (AHWJ2023A10076); Scientific Research Program of Fuyang Municipal Health Commission (FY2021-126).

**Competing interests:** The authors have declared that no competing interests exist.

GTF2E2-knockdown group. Knockdown of GTF2E2 also increased the intracellular levels of $Fe^{2+}$, lipid peroxides (LPOs) and reactive oxygen species (ROS).

## Conclusions

Our findings underscore GTF2E2's multifaceted roles in cancer biology, highlighting its potential as a diagnostic biomarker, prognostic indicator, and immunotherapeutic target across various malignancies. This investigation has the potential to contribute significantly to a deeper understanding of the substantial involvement of GTF2E2 in human malignancies, particularly UCEC.

## Introduction

Cancer remains a leading cause of death globally, significantly impacting life quality worldwide [1,2]. Despite extensive research efforts, a definitive cure for cancer has yet to be discovered [3]. The incidence and mortality rates of cancer continue to rise sharply across various populations [4,5]. The development of treatments such as radiotherapy, chemotherapy, targeted therapy, and immunotherapy represents significant progress in cancer management. These advancements underscore the ongoing efforts to better understand the complex processes of tumorigenesis. Furthermore, the expansion and refinement of public databases, such as The Cancer Genome Atlas, have facilitated the identification of new targets for immunotherapy. Adopting a comprehensive pancancer approach allows researchers to explore the connections between different cancer types, offering a unique perspective on pancancer challenges and potential strategies [6–8].

Transcription initiation factor IIE subunit beta (GTF2E2) is a crucial component of the RNA polymerase II transcription initiation complex. This protein recruits TFIIH, which is essential for clearing the promoter region, facilitating transcription by RNA polymerase II [9,10]. GTF2E2 is specifically involved in assembling the initiation complex by recruiting TFIIH, which then activates the complex through its DNA-dependent ATPase and kinase activities that target the C-terminal domain of RNA polymerase II[11,12]. Furthermore, GTF2E2 regulates the formation of the GTF2E1-GTF2E2 protein complex by controlling its protein stability through post-transcriptional mechanisms such as protein degradation [13,14]. Previous research suggested that GTF2E2 mutation is correlated with remarkable DNA repair independent transcription defects and tissue-specific dysfunction[10]. Recent bioinformatics analyses have shed light on the role of GTF2E2 in cancer progression. Yang *et al.*[15] bioinformatic study suggested that GTF2E2 promotes the development of glioblastoma by upregulating the expression of the cell division cycle 20 (CDC20). Moreover, interactions between GTF2E2 and ribosomal protein S4, X-linked (RPS4X) have been implicated in the onset of lung adenocarcinoma [16]. Recent findings also highlight the overexpression of GTF2E2 in glioma and suggest a potential link between its expression levels and the prognosis of glioma patients, indicating its involvement in cancer development [17].

In recent years, the discovery of ferroptosis, a novel mode of cell death, has opened up new perspectives in our understanding of cell fate regulation. Ferroptosis, an iron-dependent programmed cell death, is characterised by uncontrolled intracellular lipid peroxidation and an imbalance in the metabolism of iron ions. This process involves a complex regulatory network of gene expression, including changes in the expression of antioxidant genes, iron metabolism-related genes, and lipid metabolism genes [18,19]. Ferroptosis is precisely regulated at multiple levels, including epigenetic, transcriptional, posttranscriptional and

posttranslational layers [20]. For example, certain transcription factors (e.g., Nrf2 and p53) play a crucial role in the regulation of ferroptosis [21,22]. Considering the central position of GTF2E2 in transcriptional initiation, it is reasonable to speculate that it may play an important role in the process of iron death by regulating the expression of certain key genes.

In this study, we explored the expression pattern of GTF2E2 and its potential as a diagnostic and prognostic biomarker for various cancers through extensive bioinformatics analysis. In addition, we verified the biological significance of GTF2E2 in UCEC cells by in vitro experiments. Our findings suggest that GTF2E2 may serve as an important biomarker for diagnosis and prediction of prognosis in a variety of cancers. Importantly, this study preliminarily demonstrated that knockdown of GTF2E2 inhibited UCEC progression by affecting the expression of GPX4 and ACSL4 and inducing ferroptosis.

## Materials and methods

### Data processing and differential expression analysis

In our study, we analyzed mRNA expression profiles and clinical data from The Cancer Genome Atlas (TCGA), which included 33 cancer types and 18,102 samples comprising both tumor and adjacent normal tissues. We identified differentially expressed genes (DEGs) using log2 transformation and t-tests, setting a significance threshold of $P < 0.05$. We focused on genes that were consistently differentially expressed across multiple cancers. Additionally, we utilized gene expression data from 31 tissue types available through the Genotype-Tissue Expression (GTEx) project. We excluded any samples that reported zero gene expression and maintained paired samples for comparative analysis. The RNA sequencing data, initially presented in Fragments Per Kilobase per Million (FPKM), were converted to transcripts per million (TPM) using the Toil framework and further $\log_2$-transformed for detailed analysis. Statistical analyses were conducted using R software (version 4.2.1). To illustrate the expression patterns of the GTF2E2 gene across various cancers, we employed the ggplot2 package (version 3.3.6) to create bar graphs. Expression cutoffs were determined by the median expression method, and differences between groups were evaluated using the Wilcoxon rank-sum test.

### Immunohistochemistry staining of GTF2E2

The Human Protein Atlas (HPA) is a comprehensive resource detailing the distribution of protteins in various human tissues and cells. For our study, we focused on the protein expression of GTF2E2. We utilized immunohistochemical (IHC) images from the HPA, selecting twelve types of cancerous tissues and their normal counterparts. The specific cancers examined include BLCA, BRCA, CESC, COAD, GBM, HNSC, LIHC, LUAD, UCEC. This approach allowed us to assess and compare the differential expression of GTF2E2 across these diverse conditions.

### Analysis of the diagnostic value of GTF2E2

In assessing the diagnostic potential of GTF2E2 for 33 different cancer types, ROC curves were utilized. These curves were constructed based on the mRNA expression levels of GTF2E2 from both cancerous tissues and corresponding normal tissues, using data from the TCGA and GTEx databases. The analysis involved the "pROC" package (version 1.18.0) in R software for calculating the ROC curves, and the "ggplot2" package for their graphical representation. The AUC value serves as a critical indicator of diagnostic accuracy. A value closer to 1 implies higher accuracy: an AUC between 0.5 and 0.7 suggests low accuracy, between 0.7 and 0.9 indicates good accuracy, and 0.9 or above denotes high accuracy.

## Analysis of the relationships between GTF2E2, prognosis, and clinical phenotype

For the study, Kaplan-Meier analysis, using the "survival" package, assessed the overall survival (OS), disease-specific survival (DSS), and progression-free interval (PFI) across 33 cancer types, comparing groups with high and low expressions of the GTF2E2 gene. Cox regression analysis provided the $P$ values. The results, including hazard ratios (HR) and their 95% confidence intervals, were visually depicted in forest plots using the "survminer" and "ggplot2" packages. Additionally, we explored the relationship between GTF2E2 expression and two clinical phenotypes—tumor stage and patient age—using the "limma" and "ggpubr" R packages, defining statistical significance at $P < 0.05$.

## GTF2E2 expression in different molecular and immune subtypes of cancers

To investigate the association between GTF2E2 expression and various molecular and immune subtypes across 33 cancer types, we employed the "subtype" module of the TISIDB database. TISIDB serves as an integrated platform that collates data to explore the interactions between tumor biology and immune responses [23]. We specifically analyzed GTF2E2 mRNA expression across six immune subtypes: C1 (wound healing), C2 (IFN-γ dominant), C3 (inflammatory), C4 (lymphocyte depleted), C5 (immunologically quiet), and C6 (TGF-β dominant). This analysis enabled a detailed examination of the correlation between GTF2E2 expression levels and immune response patterns in cancers.

## Genetic alteration analysis of GTF2E2

To analyze genetic alterations in the GTF2E2 gene, we utilized the cBioPortal as our primary data source, referencing studies from the TCGA PanCancer Atlas [24,25]. We examined the frequency of somatic mutations and detailed genomic characteristics of GTF2E2 mutations across different cancer types. This analysis involved using the "Cancer Type Summary," "Mutations," and "mRNA vs. Study" modules on the portal, allowing us to identify specific mutation sites within the gene.

## PPI network analyses of GTF2E2

In our study, we used the STRING database [26] to gather and analyze potential protein interactions related to GTF2E2. This data was integrated into a protein-protein interaction (PPI) network, where we applied a significance threshold of a confidence scores above 0.7. For visualization and further examination, the data was transferred to Cytoscape (version 3.8.0). Within Cytoscape, we utilized the cytoHubba plugins to pinpoint key network modules. We identified the top 10 central nodes, ranked by the MCC method, which were designated as hub genes. To delve deeper into the functional pathways, we employed the PathLinker plugins to map out pathways connected to these hub genes. Additionally, we explored the correlations between these hub genes in various cancer types to understand their potential roles in cancer biology more comprehensively. This analysis aids in identifying critical components and interactions that could serve as diagnostic or therapeutic targets in cancer.

## Functional enrichment analysis of GTF2E2

In our study, we analyzed the gene functions and pathway associations related to GTF2E2 using Gene Ontology (GO) and Kyoto Encyclopedia of Genes and Genomes (KEGG) enrichment analyses. To identify genes that interact closely with GTF2E2, we utilized data from the STRING database and analyzed it with the "clusterProfiler" and "org.Hs.e.g.,db" packages in

R. We set a significance threshold of $P < 0.01$ for identifying relevant GO functions and KEGG pathways. The results of these analyses were presented in bubble charts created with the "ggplot2" package.

### Gene set enrichment analysis

In this study, we used the "clusterProfiler" package for gene set enrichment analysis (GSEA) to explore the biological pathway differences between groups with high and low levels of GTF2E2. We identified significant pathway alterations using a false discovery rate (FDR) < 0.25 and an adjusted $P < 0.05$. The analysis included 1,000 gene set permutations. The top five results from this analysis were displayed using mountain plots, created with the "ggplot2" package in R, ensuring clear visual representation of the findings.

### Comprehensive analysis of GTF2E2's role in cancer genomics

GSCALite is a robust platform for analyzing cancer genomics, integrating data from several sources. This includes genomic data from TCGA, which covers 33 cancer types, drug response data from GDSC and CTRP, and normal tissue information from GTEx [27]. Our analysis focused on GTF2E2, a gene associated with cancer, across these varied cancer types using this integrated platform. We examined major cancer-related pathways to see whether they are activated or inhibited by GTF2E2. These pathways include TSC/mTOR, receptor tyrosine kinases, RAS/MAPK, PI3K/AKT, as well as hormone estrogen receptor (ER) and hormone androgen receptor (AR). Other pathways analyzed include epithelial–mesenchymal transition (EMT), DNA damage response (DDR), cell cycle regulation, and apoptosis. This comprehensive analysis helps understand the molecular mechanisms where GTF2E2 might influence cancer progression and treatment responses.

### CancerSEA

To examine the role of GTF2E2 across various cancers, we utilized the CancerSEA database, which provides insights into cancer cell functionality at the single-cell level [28]. Our analysis focused on determining how GTF2E2 correlates with several functional states, including invasion, metastasis, proliferation, EMT, angiogenesis, apoptosis, and others, across seven cancer types. We established a correlation threshold of 0.3 and required a significance level of $P < 0.05$ to assess the relationships between GTF2E2 expression and these functional states in each type of cancer.

### Relationship between GTF2E2 expression and immunity

In our study, we utilized the "GSVA" package and the "ssGSEA" algorithm to examine how GTF2E2 expression correlates with immune-related factors. This includes tumor-infiltrating lymphocytes, immunostimulators, immunoinhibitors, MHC molecules, and chemokines and their receptors, across 33 cancer types. We used Spearman's correlation analysis to measure the strength of these correlations, considering results statistically significant at $P < 0.05$.

### Interaction of GTF2E2 with chemicals and genes

The Comparative Toxicogenomics Database (CTD) is an online tool that helps researchers' study how chemicals affect health by uncovering new connections at the molecular level [29]. In our study, we utilized the CTD to identify chemicals that interact with the gene GTF2E2 and to find genes similar to GTF2E2 based on their chemical interactions. This approach has provided valuable insights into the potential mechanisms through which GTF2E2 influences cancer development.

## Cell culture

Human UCEC cell lines (ISK and HEC-1-A) were sourced from the NHC Key Laboratory of Abnormal Gametes and Reproductive Tract at Anhui Medical University. The cells were cultured in DMEM medium supplemented with 10% fetal bovine serum and 1% cyan-stranded biclonal antibody. Cultures were maintained at 37°C with 5% $CO_2$, with the medium refreshed every two days. Once the cells reached 80% confluence, they were passaged for further experiments.

## CCK8 assay

The CCK8 assay was used to assess cell proliferation. ISK and HEC-1-A cells were seeded into 96-well plates at a density of 2000 cells per well. After 24 hours of incubation (considered 0 h), the cells were transfected. At 24-, 48-, and 72-hours post-transfection, 10 μl of CCK8 solution was added to each well, and the cells were incubated at 37°C for 2 hours. The optical density (OD) at 450 nm was then measured using a microplate reader to calculate cell viability.

## Transwell

Initially, 50,000 cells were seeded into the upper chamber with 200 μl of medium containing 1% serum, while 700 μl of medium with 10% serum was added to the lower chamber. After 48 hours, the cells in the upper chamber were removed, and the remaining cells in the lower chamber were fixed with 4% paraformaldehyde for 20 minutes. The cells were then stained with 0.1% crystal violet for 30 minutes. Images were captured using an Olympus microscope.

## Western blot analysis

ISK and HEC-1-A cells were lysed with precooled RIPA buffer to extract total protein (n ≥ 3). The protein samples were separated by SDS-PAGE and transferred to a PVDF membrane. After blocking with 5% skim milk for 1 hour, the membranes were sectioned and incubated overnight at 4°C with primary antibodies. The following day, they were treated with anti-rabbit or anti-mouse IgG secondary antibodies (1:10,000) for 1 hour. Chemiluminescence detection was performed using a CS analysis system (5200, Tanon, Shanghai, China). The antibodies used included anti-GTF2E2, anti-ACSL4, anti-GPX4, and anti-beta actin (all at 1:1000 dilution). Protein bands were quantified using ImageJ software and normalized to control values.

## Cell transfection

The GTF2E2 siRNA sequences were purchased from Sangon Biotech, and the sequences were as follows: GTF2E2-siRNA1: forwards 5′- GCAUGACCAGCGAGGAUUA-3′; reverse, 5′-UAAUCCUCGCUGGUCAUGC -3′. GTF2E2-siRNA2: forwards 5′- AGUUUGGUGUUC UUGCUAA-3′; reverse, 5′-UUAGCAAGAACACCAAACU -3′.Briefly, ISK and HEC-1-A cells were transiently transfected with GTF2E2 siRNA or negative control siRNA when they reached 50–80% confluence according to the manufacturer's protocol, and the transfection efficiency was assessed by WB 48 h later.

## Measurement of reactive oxygen species

Reactive oxygen species (ROS) levels were measured using a ROS Assay Kit. ISK and HEC-1-A cells were seeded in 6-well plates at 80–90% confluence, following the kit instructions. After treatment with either siNC or GTF2E2-siRNA, the cells were washed twice with PBS. A 1:1000 dilution of the ROS probe in serum-free medium was added, and the cells were

incubated at 37°C for 20 minutes. After two to three washes with serum-free medium, ROS levels and fluorescence intensity were analyzed using an Olympus microscope (Olympus IX73, Japan).

### Lipid peroxidation assay

C11 BODIPY 581/591 was dissolved in DMSO to 1 mM for storage. To assess intracellular lipid peroxides (LPOs), ISK cells or HEC-1-A cells were seeded in 6-well plates. After treatment with siNC or GTF2E2-siRNA, the cells were incubated with 1 ml of serum-free medium containing 2 μM BODIPY 581/591C11 dye for 30 min in a 37°C incubator and washed two to three times with PBS. Fluorescence intensity was detected using an Olympus microscope (Olympus IX73, Japan).

### Determination of $Fe^{2+}$ levels

ISK cells or HEC-1-A cells were inoculated into 6-well plates at a density of 80–90% according to the instructions. After treatment with siNC or GTF2E2-siRNA, cells were washed twice with PBS. The FeRhoNox-1 probe was then diluted 1:1000 in serum-free medium and added to the cells, which were incubated for 30 minutes at 37°C in an incubator and washed two to three times with serum-free medium. $Fe^{2+}$ content and fluorescence intensity were detected using an Olympus microscope (Olympus IX73, Japan).

### Reagents and antibodies

The Reactive Oxygen Species Assay Kit (S0033S) and Lipid Peroxidation Assay Kit with BODIPY 581/591 C11 (S0043M) were purchased from Beyotime (Shanghai, China). A Cell Counting Kit-8 (CT0001) was purchased from SparkJade (Shandong, China). FerroOrange live cell probe (MX4559-24UG) was purchased from Maokang Biotechnology (Shanghai, China). The primary antibodies used in this study were as follows: anti-GPX4 (381958), anti-beta actin (250136), goat anti-mouse IgG H&L (HRP) (511103) and goat anti-rabbit IgG H&L (HRP) (511203), which were purchased from Zen BioScience (Chengdu, China); The anti-ACSL4 antibody (22401–1-AP) was purchased from Proteintech (USA). The anti-GTF2E2 (TFIIE-β) antibody (BD-PT4616) was purchased from Biodragon (Jiangsu, China). Malondialdehyde (MDA) content kit (#JL-T0761) was purchased from Jianglai (Shanghai, China).

### Statistical analysis

Statistical analysis was performed using SPSS 26.0 and GraphPad Prism. Quantitative data were expressed as mean ± standard deviation (Mean ± SD). For continuous variables with a normal distribution and equal variance (as confirmed by the chi-square test), an independent samples t-test or ANOVA was used. For non-normally distributed continuous variables, the rank-sum test was applied. Categorical variables, presented as frequencies (percentages), were analyzed using the chi-square test. $P < 0.05$ was considered statistically significant.

## Results

### Differential expression of GTF2E2 between tumor and normal tissue samples

GTF2E2 mRNA and protein are broadly expressed across a range of organs and tissues (Fig 1A). Our analysis incorporates data from the HPA, with 375 normal tissue samples, and the GTEx project, with 13,084 samples. The mRNA of GTF2E2 is notably prevalent in the pancreas, testis, ovary, adrenal gland, liver, tonsil, esophagus, lymph node, cervix, and stomach

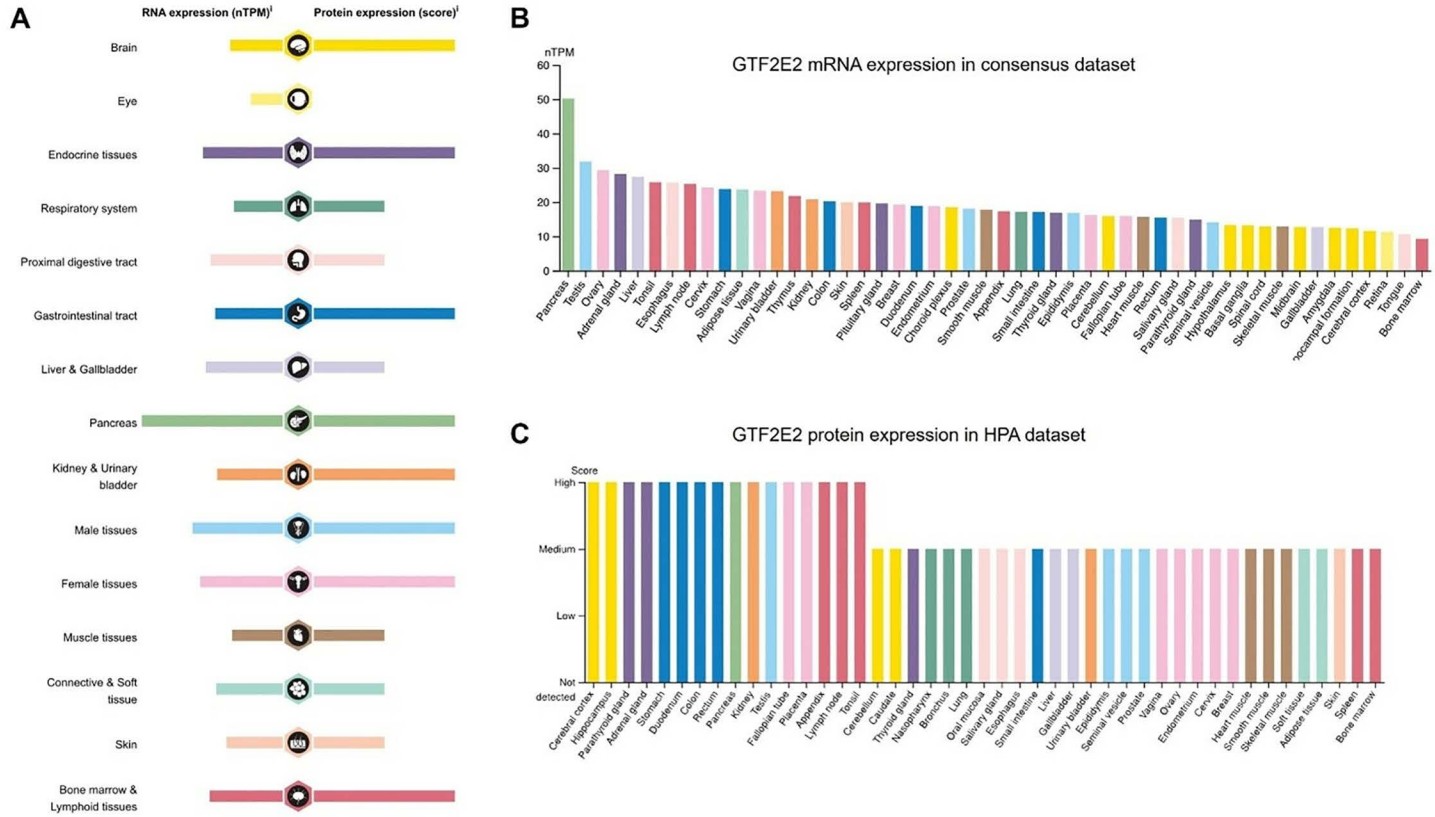

**Fig. 1. RNA and protein expression profile of GTF2E2 in human organs and tissues.** (A) The summary of GTF2E2 mRNA and protein expression in human organs and tissues; (B) GTF2E2 mRNA expression summary in different human organs and tissues based on consensus dataset; (C) GTF2E2 protein expression summary in different human organs and tissues.

(Fig 1B). Protein data from the HPA, covering 144 individuals across 44 tissue types, shows significant presence in the cerebral cortex, hippocampus, parathyroid gland, adrenal gland, stomach, duodenum, colon, rectum, pancreas, and kidney (Fig 1C). We evaluated GTF2E2 mRNA expression in 33 types of cancer (Table 1), analyzing 18,102 unpaired samples (Fig 2A). displays the inclusion of 18,102 samples in the unpaired sample analysis. In comparison to normal samples, low GTF2E2 mRNA expression was noted in LAML, OV, TGCT, SKCM, PRAD (all $P < 0.001$), KICH ($P = 0.015$), LIHC ($P = 0.043$) and PCPG ($P = 0.004$), while high expression was observed in ACC, BLCA, CHOL, COAD, DLBC, GBM, HNSC, KIRC, KIRP, LUAD, LUSC, READ, STAD, THCA, THYM, UCEC, UCS (all $P < 0.001$), and CESC ($P = 0.008$). Some cancers, such as MESO and UVM, could not be analyzed due to a lack of normal tissue comparisons. Compared to paracancerous tissue, GTF2E2 mRNA expression was significantly higher in BLCA, CHOL, COAD, ESCA, GBM, HNSC, KIRC, KIRP, LUAD, LUSC, STAD, THCA, UCEC (all $P < 0.001$), BRCA ($P = 0.029$), LIHC ($P = 0.015$) and READ ($P = 0.033$). This paired analysis involved 11,123 samples (Fig 2B). ACC, DLBC, LAML, LGG, MESO, OV, TGCT, UCS, and UVM could not be analyzed due to insufficient paracancerous samples. No differences were observed in CESC, KICH, PAAD, PRAD, SARC, SKCM, and THYM ($P > 0.05$). Among the paired sample analyses, which involved 1404 samples in 23 cancers and 1404 paracancerous samples, GTF2E2 mRNA expression increased significantly in CHOL, COAD, HNSC, KIRC, KIRP, LUAD, LUSC, THCA, UCEC (all $P < 0.001$), BLCA ($P = 0.014$), ESCA ($P = 0.018$), PRAD ($P = 0.010$), READ ($P = 0.007$), and STAD ($P = 0.020$) (Fig 2C).

**Table 1. TCGA cancer abbreviations and the corresponding cancer type.**

| Abbreviations | Cancer type |
|---|---|
| ACC | Adrenocortical carcinoma |
| BLCA | Bladder urothelial carcinoma |
| BRCA | Breast invasive carcinoma |
| CESC | Cervical squamous cell carcinoma and endocervical adenocarcinoma |
| CHOL | Cholangiocarcinoma |
| COAD | Colon adenocarcinoma |
| DLBC | Lymphoid neoplasm diffuses large B-cell lymphoma |
| ESCA | Esophageal carcinoma |
| GBM | Glioblastoma multiforme |
| HNSC | Head and neck squamous cell carcinoma |
| KICH | Kidney chromophobe |
| KIRC | Kidney renal clear cell carcinoma |
| KIRP | Kidney renal papillary cell carcinoma |
| LAML | Acute myeloid leukemia |
| LGG | Brain lower grade glioma |
| LIHC | Liver hepatocellular carcinoma |
| LUAD | Lung adenocarcinoma |
| LUSC | Lung squamous cell carcinoma |
| MESO | Mesothelioma |
| OV | Ovarian serous cystadenocarcinoma |
| PAAD | Pancreatic adenocarcinoma |
| PCPG | Pheochromocytoma and paraganglioma |
| PRAD | Prostate adenocarcinoma |
| READ | Rectum adenocarcinoma |
| SARC | Sarcoma |
| SKCM | Skin cutaneous melanoma |
| STAD | Stomach adenocarcinoma |
| TGCT | Testicular germ cell tumors |
| THCA | Thyroid carcinoma |
| THYM | Thymoma |
| UCEC | Uterine corpus endometrial carcinoma |
| UCS | Uterine carcinosarcoma |
| UVM | Uveal melanoma |

At the protein level, we reviewed IHC results from the HPA database and compared these to gene expression data from TCGA. Our findings, depicted across (Fig 3A–3I), indicate that protein expression of GTF2E2 is substantially higher in 9 types of cancers compared to normal tissues. This consistency between mRNA and protein data underscores the potential diagnostic and therapeutic relevance of GTF2E2 in cancer.

## Diagnostic value of GTF2E2 across cancers

Fig 4A-T highlights the substantial diagnostic potential of GTF2E2 across a spectrum of cancers. Its AUC was greater than 0.6 in 24 cancers and even exceeded 0.7 in 20 cancers, including BLCA (AUC = 0.741), CESC (AUC = 0.736), CHOL (AUC = 0.946), COAD (AUC = 0.857), DLBC (AUC = 0.841), ESCA (AUC = 0.819), GBM (AUC = 0.966), HNSC (AUC = 0.777), KIRP (AUC = 0.909), KIRC (AUC = 0.770), LAML (AUC = 0.993), LUAD

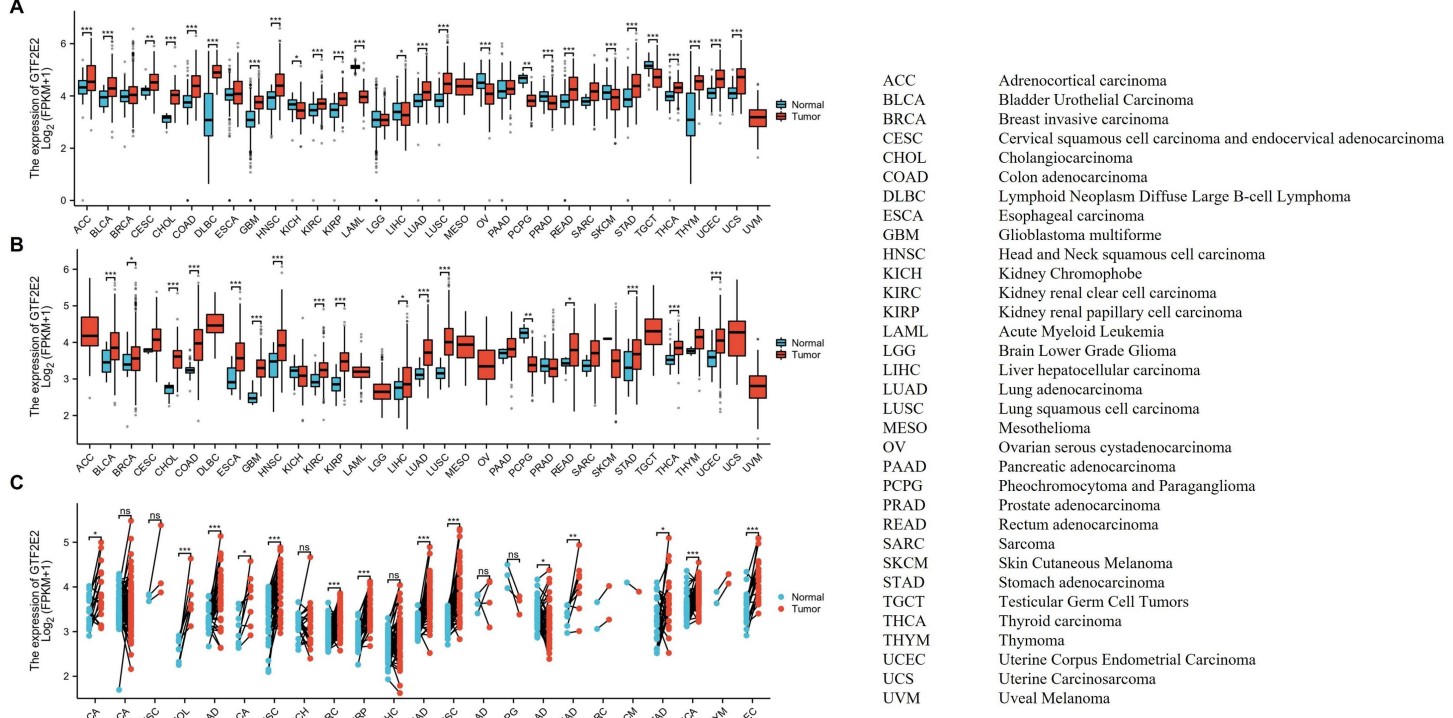

**Fig. 2. The expression of GTF2E2 mRNA in pan-cancer.** (A) Expression of GTF2E2 between the 33 cancers and normal tissues in unpaired sample analysis; (B) Expression of GTF2E2 between the 33 cancers and paracancerous tissues in unpaired sample analysis; (C) Paired sample analysis of GTF2E2 mRNA expression between 23 cancers and paracancerous tissues in BLCA, BRCA, CESC, CHOL, COAD, ESCA, HNSC, KICH, KIRC, KIRP, LIHC, LUAD, LUSC, PAAD, PCPG, PRAD, READ, SARC, SKCM, STAD, THCA, THYM and UCEC. $^{*}P < 0.05$, $^{**}P < 0.01$, $^{***}P < 0.001$. ns, Not Significant.

(AUC = 0.880), LUSC (AUC = 0.946), OSCC (AUC = 0.778), OV (AUC = 0.747), PCPG (AUC = 0.984), SARC (AUC = 0.719), THCA (AUC = 0.787), THYM (AUC = 0.775) and UCEC (AUC = 0.779) (Fig S1), which had high diagnostic value.

## Prognostic value of GTF2E2 across cancers

We performed Kaplan-Meier analysis to assess the prognostic value of GTF2E2. Cox regression analysis revealed a significant association between GTF2E2 expression and OS in 15 out of 33 cancers (Fig 5). The results showed that low GTF2E2 expression was associated with significantly better OS in ACC, ESAD, GBM, KICH, KIRC, KIRP, LGG, LIHC, LUAD, MESO, PAAD and UVM. In contrast, high GTF2E2 expression was linked to significantly better OS in CESC, SKCM and THCA (Fig 6A–6O). In terms of DSS among these cancers, GTF2E2 had a protective effect in CESC, LUSC and SKCM but was associated with increased risk in ACC, ESAD, GBM, LGG, KICH, KIRC, KIRP, LUAD, MESO, UVM and OSCC (Fig 5, Fig S2A–S2N). In the analysis of the PFI across these cancers, GTF2E2 was associated with an increased risk in ACC, LGG, KICH, KIRC, KIRP, LIHC, MESO and PAAD (Fig 5, Fig S3A–S3L).

## GTF2E2 expression in different immune and molecular subtypes across cancers

Based on previous research, we discovered that GTF2E2 expression levels significantly affect OS across 15 different types of cancer. To deepen our understanding, we extended our analysis to both the immune and molecular subtypes of these cancers, as well as exploring

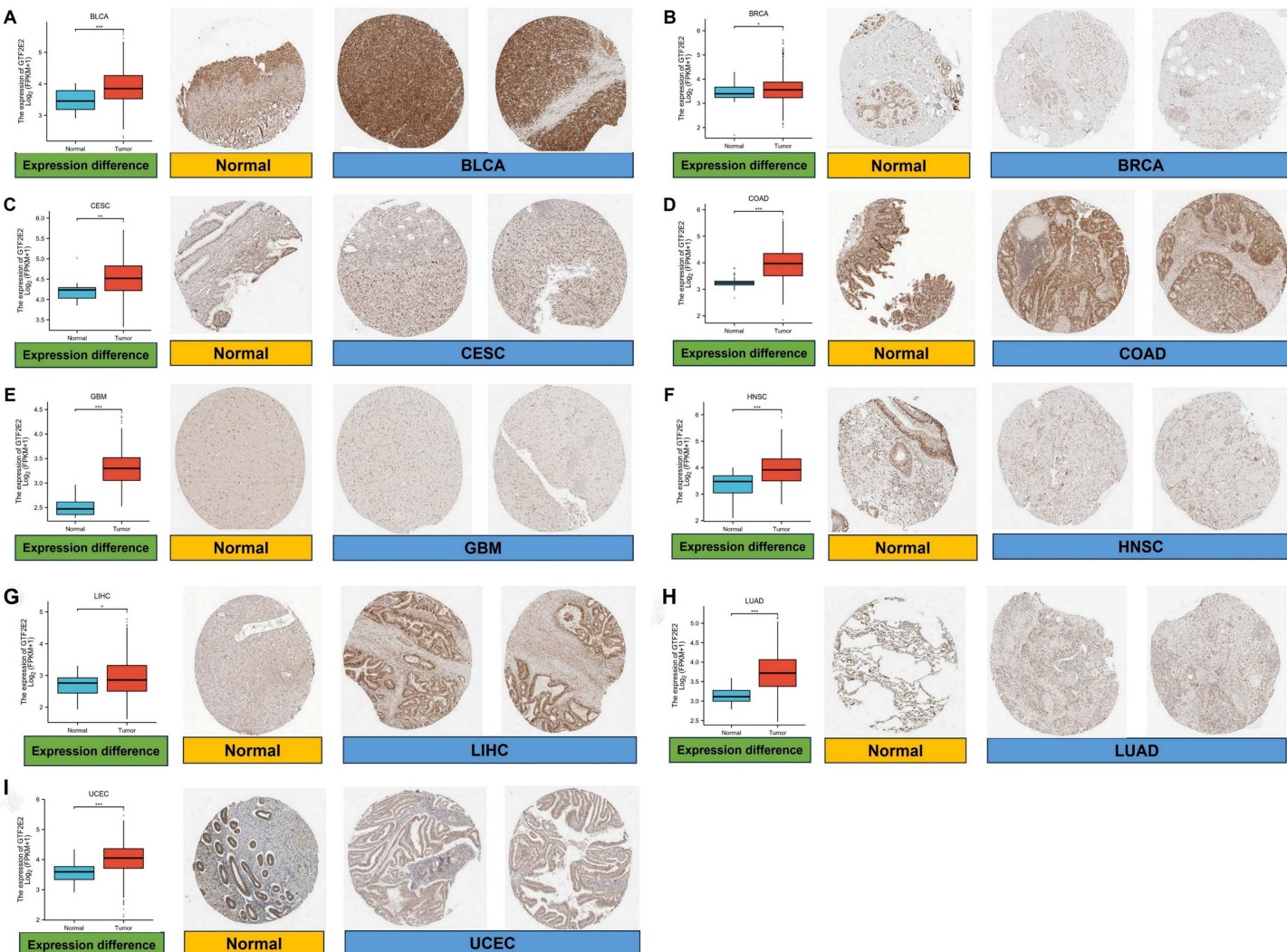

**Fig. 3. Comparison of GTF2E2 gene expression between normal and tumor tissues (left) and immunohistochemistry images in normal (middle) and tumor (right) tissues.** $^*P < 0.05$, $^{**}P < 0.01$, $^{***}P < 0.001$.

an additional 18 types of cancer. We first examined the immune subtypes of the original 15 cancers and found substantial variations in GTF2E2 expression in seven cancers: PAAD across five subtypes; LGG and LUAD, each in four subtypes; GBM in three subtypes; and KIRC, KIRP, and MESO, each in six subtypes (Fig 7A–7G). Further analysis included 18 additional cancer types. Significant variations in GTF2E2 expression were observed within immune subtypes of BRCA, COAD, ESCA, LUSC, PRAD, READ, SARC, STAD, TGCT, and UCEC (Fig S4A–S4J). Regarding molecular subtypes, notable differences in GTF2E2 expression were identified in 11 cancer types, including BRCA, COAD, HNSC, KIRP, LGG, LIHC, PCPG, PRAD, READ, STAD, and UCEC (Fig 8A–8K).

### Correlation of GTF2E2 expression with clinical phenotypes in various cancers

In our analysis of tumor stage relevance, we observed that GTF2E2 expression significantly increased at early stages in 12 cancer types, including BLCA, CHOL, COAD, ESCA, HNSC, KIRC, KIRP, LUAD, LUSC, OSCC, STAD, and THCA (Fig 9). This elevation in GTF2E2

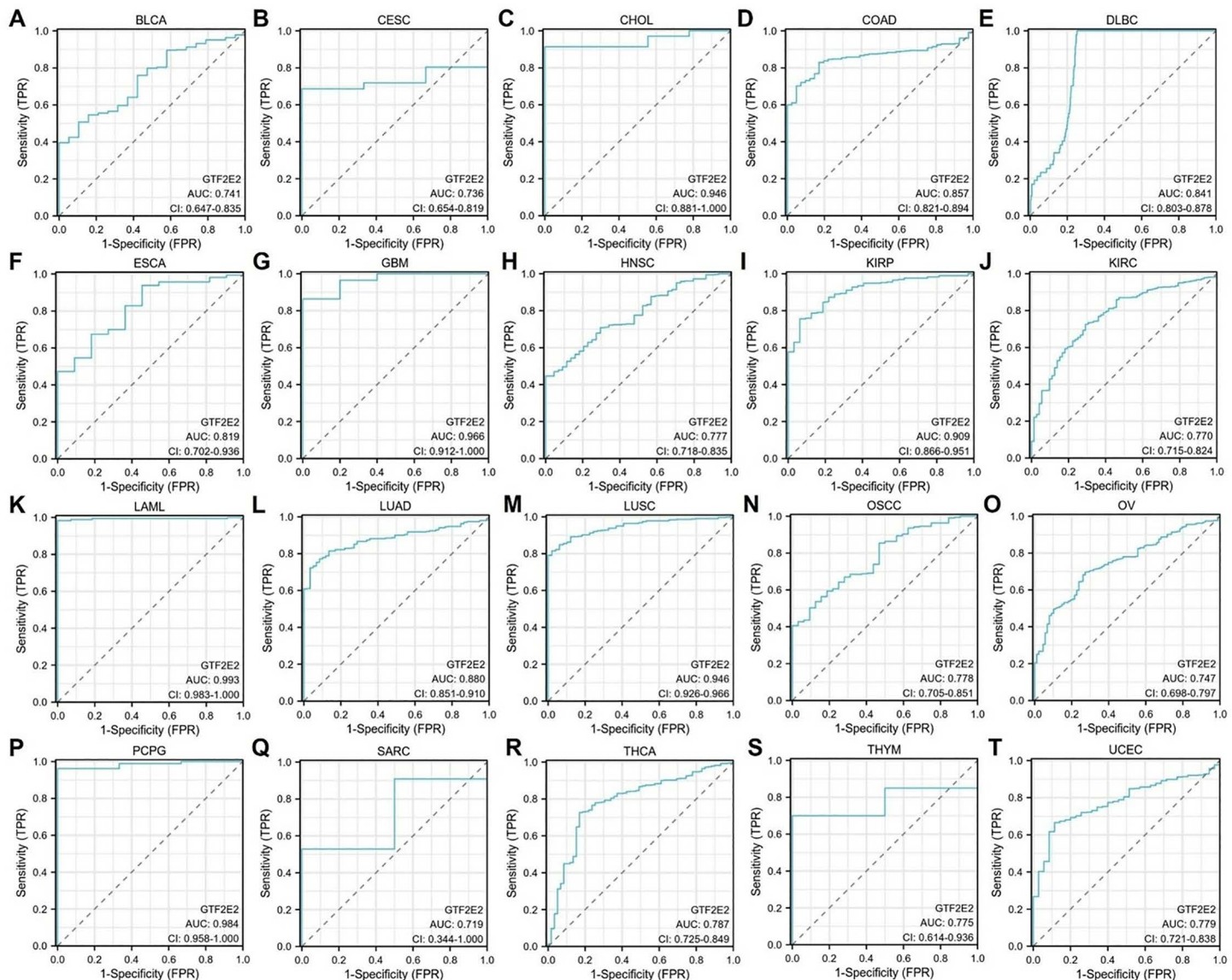

**Fig. 4. Receiver Operator Characteristic (ROC) curve of GTF2E2 in 20 Cancers.** Cancers with AUC > 0.7 for GTF2E2: (A) BLCA, (B) CESC, (C) CHOL, (D) COAD, (E) DLBC, (F) ESCA, (G) GBM, (H) HNSC, (I) KIRP, (J) KIRC, (K) LAML, (L) LUAD, (M) LUSC, (N) OSCC, (O) OV, (P) PCPG. (Q) SARC, (R) THCA, (S) THYM, (T) UCEC.

expression suggests its potential value in early cancer diagnosis. Subsequently, we investigated the variation in GTF2E2 expression based on patient age across different cancers. Notably, patients aged 60 years or younger exhibited higher GTF2E2 levels in BRCA (Fig 10A; $P$ = 0.001), LIHC (Fig 10C; $P$ < 0.001), THYM (Fig 10F; $P$ = 0.002), and UCEC (Fig 10G; $P$ < 0.001). Patients younger than 65 years with LUAD also showed higher expression (Fig 10D; $P$ = 0.004). Conversely, in STAD, higher GTF2E2 expression was noted in patients older than 65 years (Fig 10E; $P$ = 0.019). Additionally, patients over 40 years with LGG displayed increased expression (Fig 10B; $P$ = 0.019). No significant age-related differences in GTF2E2 expression were observed in other cancers. These findings underscore the potential of GTF2E2 as a marker for early diagnosis and its differential expression across ages in several cancers.

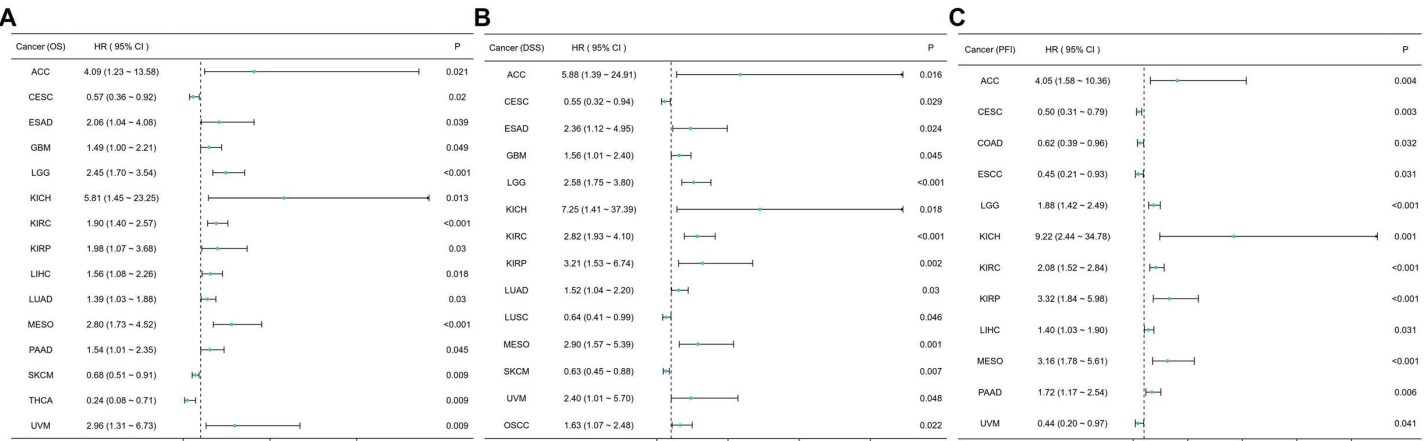

**Fig. 5. K-M analysis for high- and low- GTF2E2 gene expression in cancers.** Forest plot of GTF2E2 OS in 15 cancers (A), DSS in 14 cancers (B), and PFI in 12 cancers (C).

## Genetic alteration of GTF2E2

In our study, we utilized the cBioPortal tool to examine genetic alterations in the expression of the gene GTF2E2 across a spectrum of cancers, based on data from 32 studies within the TCGA PanCancer Atlas, comprising 10,967 samples. Our analysis identified 64 mutation sites in the GTF2E2 gene sequence, ranging from amino acid 0–291. These included 41 missense mutations, 14 truncating mutations, two splicing mutations, and seven structural variations or fusions. The mutation R194H/C emerged as the most common (Fig 11A). The predominant types of mutations observed were deep deletions, missense mutations, and gene amplifications. GTF2E2 mutations appeared most frequently in cancers of the PRAD, UCEC, BLCA, LIHC, COAD, OV, UCS, and BRCA (Fig 11B). Additionally, a shallow deletion in GTF2E2 mRNA expression was generally prevalent across the studied cancer types, except in LAML and THYM (Fig 11C).

## The PPI, functional enrichment, and gene set enrichment of GTF2E2 in cancers

We identified 50 genes closely related to GTF2E2 using the STRING database and constructed a PPI network, setting a specific threshold for inclusion (Fig 12A). The top 10 central genes in this network were TAF8, TAF2, TAF9, POLR2B, ERCC2, TAF11, TAF13, TAF5, GTF2B, and GTF2E2, as shown in (Fig 12B). These genes are notably linked in cancer types where GTF2E2 expression affects patient outcomes, with the exception of BLCA, CHOL, DLBC, PAAD, and UCS (Fig 12C). Further analysis was performed through GO and KEGG enrichment assessments to categorize the RNA functionalities of these genes into three domains: biological processes (BP), molecular functions (MF), and cellular components (CC). Significant biological processes included DNA-templated transcription initiation, transcription initiation from RNA polymerase II promoter, and positive regulation of transcription from the same promoter. The cellular components primarily involved were DNA-directed RNA polymerase complex, nuclear DNA-directed RNA polymerase complex, and the RNA polymerase II holoenzyme. Molecular functions encompassed activities related to general transcription initiation factors, RNA polymerase II general transcription initiation factors, and DNA-directed 5'-3' RNA polymerase. Key pathways identified through KEGG analysis included basal transcription factors, RNA polymerase, and nucleotide excision repair (Fig 12D). In the context of

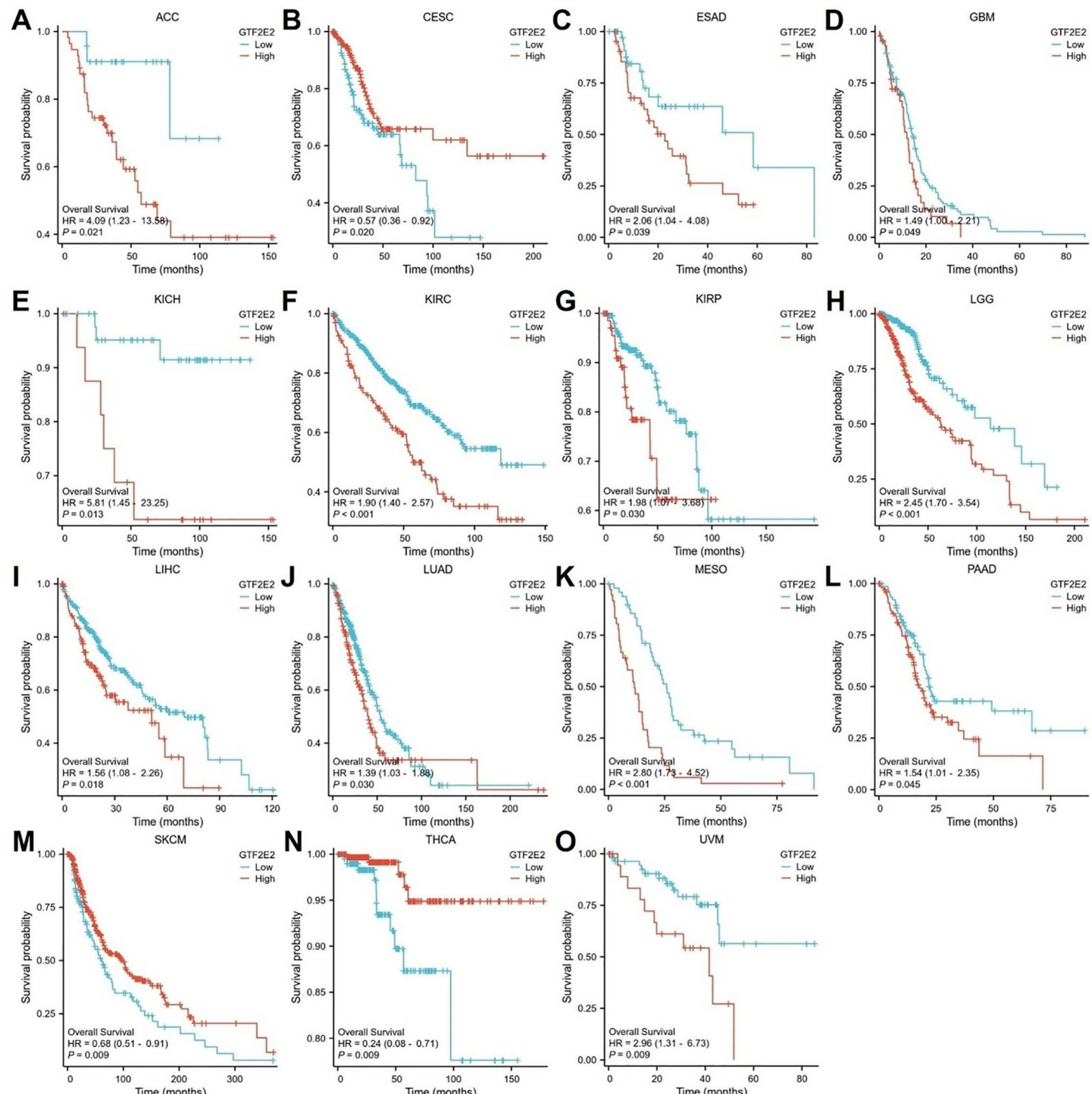

**Fig. 6. Correlations between GTF2E2 and prognosis in 15 cancers.** OS K-M curve for GTF2E2 in 15 cancers. The unit of X-axis is month. (A) ACC, (B) CESC, (C) ESAD, (D) GBM, (E) KICH, (F) KIRC, (G) KIRP, (H) LGG, (I) LIHC, (J) LUAD, (K) MESO, (L) PAAD, (M) SKCM, (N) THCA, (O) UVM.

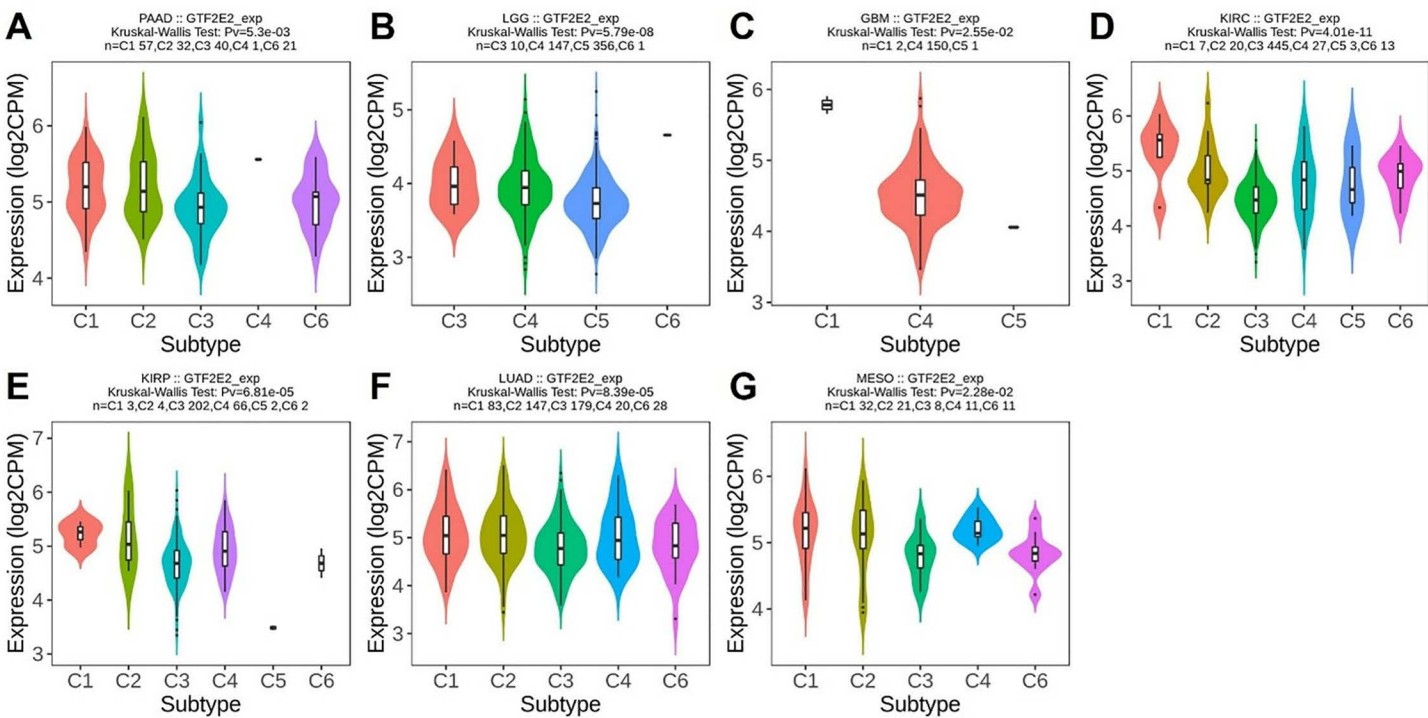

**Fig. 7. Correlations between GTF2E2 expression and immune subtypes in 7 cancers.** (A) PAAD, (B) LGG, (C) GBM, (D) KIRC, (E) KIRP, (F) LUAD, (G) MESO. C1 (wound healing), C2 (IFN-γ dominant), C3 (inflammatory), C4 (lymphocyte deplete), C5 (immunologically quiet), and C6 (TGF-β dominant).

cancer, GTF2E2 was found to activate pathways associated with apoptosis, cell cycle regulation, EMT, DDR, and AR signaling. In contrast, it inhibits pathways related to TSC/mTOR, RTK, RAS/MAPK signaling, PI3K/AKT, and ER signaling (Fig 12E). (Fig 13A–13O) displays the GSEA results for 15 different cancer types. The analysis consistently showed enrichment in several key pathways: antigen binding, T cell receptor complex, B cell receptor signaling pathway, chromosome segregation, and immunoglobulin complex. These findings highlight a strong link between GTF2E2 expression and critical biological processes, including DNA transcription, T cell activation, B cell activation, and protein synthesis, across multiple cancer contexts.

## Functional states of GTF2E2 in scRNA-seq datasets

We explored the role of the GTF2E2 gene across various cancers using the CancerSEA database to assess its impact at the single-cell level. This analysis revealed that GTF2E2 expression is generally associated with increased cell cycle activity, angiogenesis, differentiation, inflammation, metastasis, and cell proliferation. However, we found weaker negative correlations with invasion, DNA repair, DNA damage, and apoptosis (Fig 14A). Additionally, we delved into the specific cancer types to examine the relationship between GTF2E2 and their functional statuses. For example, in Acute Myeloid Leukemia (AML), GTF2E2 was positively associated with functions like differentiation, metastasis, proliferation, quiescence, inflammation, EMT, hypoxia, and angiogenesis. Similarly, in Retinoblastoma (RB), positive correlations were observed with angiogenesis, differentiation, and inflammation. In LUAD, the gene was linked predominantly to metastasis, while in High-grade Glioma (HGG), it correlated with cell cycle activity. Conversely, our results also highlighted specific negative correlations in certain

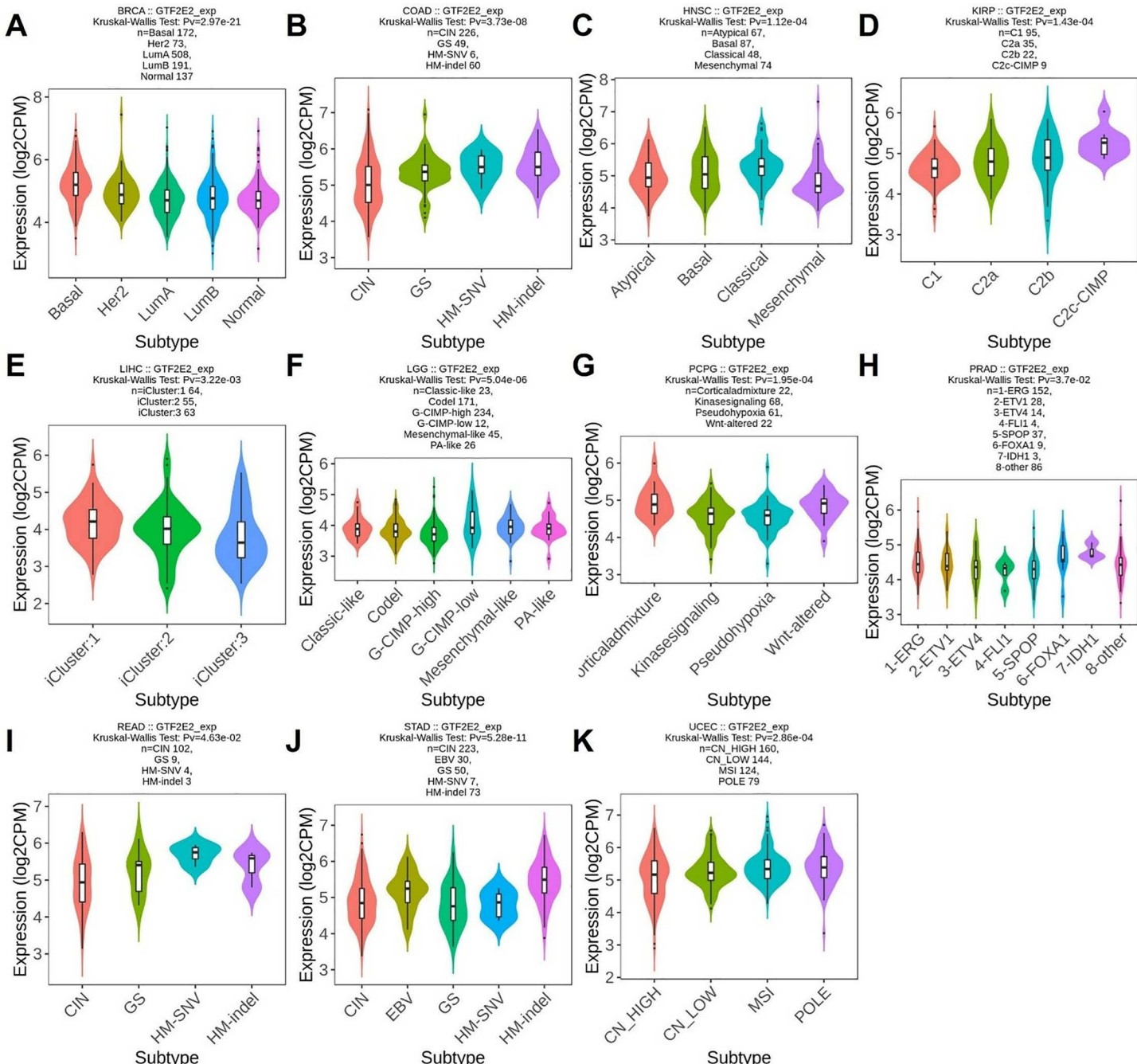

**Fig 8. Correlations between GTF2E2 expression and molecular subtypes in 11 cancers.** (A) BRCA, (B) COAD, (C) HNSC, (D) KIRP, (E) LIHC, (F) LGG, (G) PCPG, (H) PRAD, (I) READ, (J) STAD, (K) UCEC.

cancers. In AML, GTF2E2 showed negative associations with DNA repair. RB exhibited negative links with DNA repair, the cell cycle, and DNA damage. In Uveal Melanoma (UM), the gene negatively correlated with DNA repair, DNA damage, apoptosis, metastasis, and invasion. Additionally, in Colon and Rectal Cancer (CRC) as well as OV, there was a negative correlation with invasion (Fig 14B–14H).

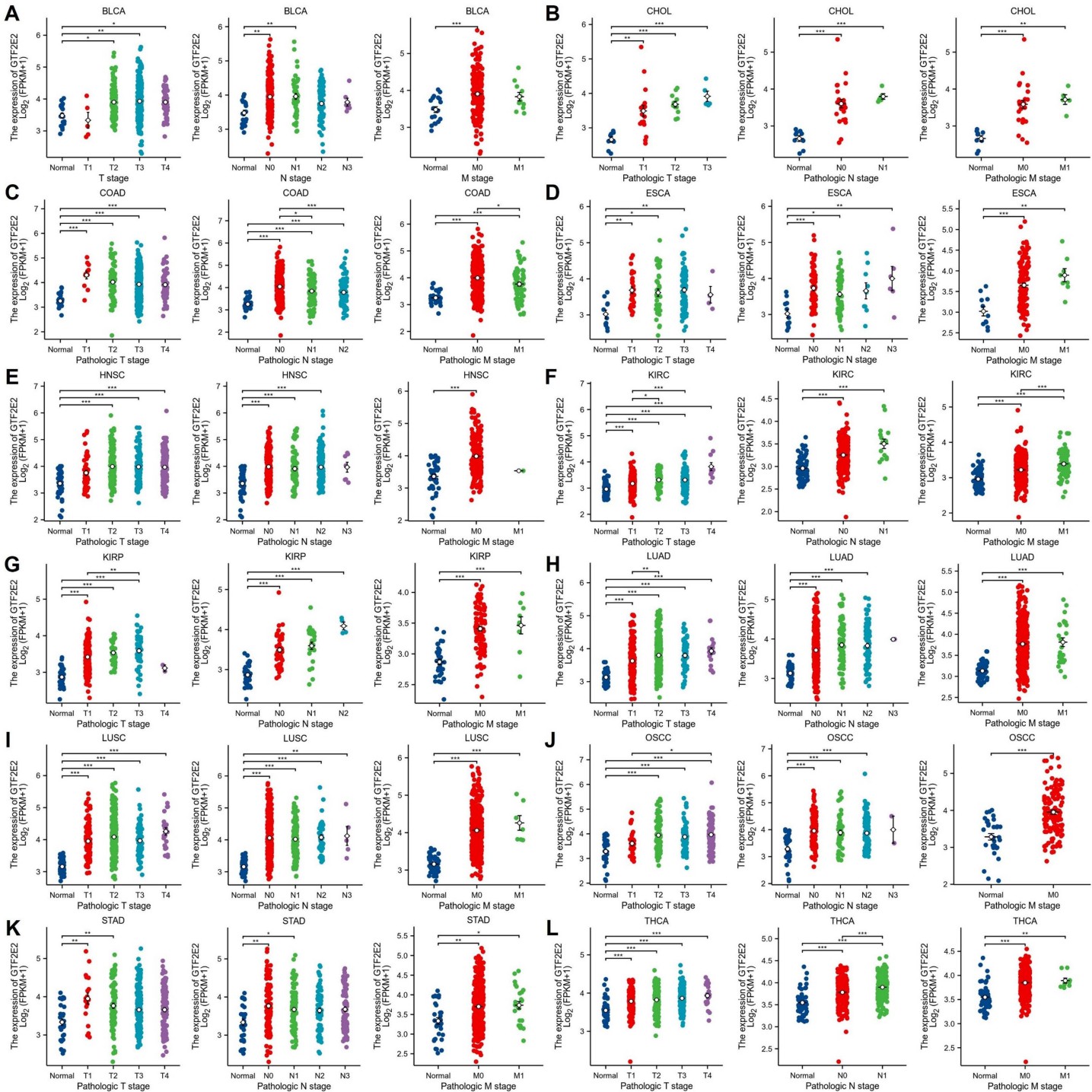

**Fig. 9. Association between GTF2E2 expression and tumor stage.** (A) BLCA, (B) CHOL, (C) COAD, (D) ESCA, (E) HNSC, (F) KIRC, (G) KIRP, (H) LUAD, (I) LUSC, (J) OSCC, (K) STAD, (L) THCA. $^*P < 0.05$, $^{**}P < 0.01$, $^{***}P < 0.001$. ns, not statistically significant.

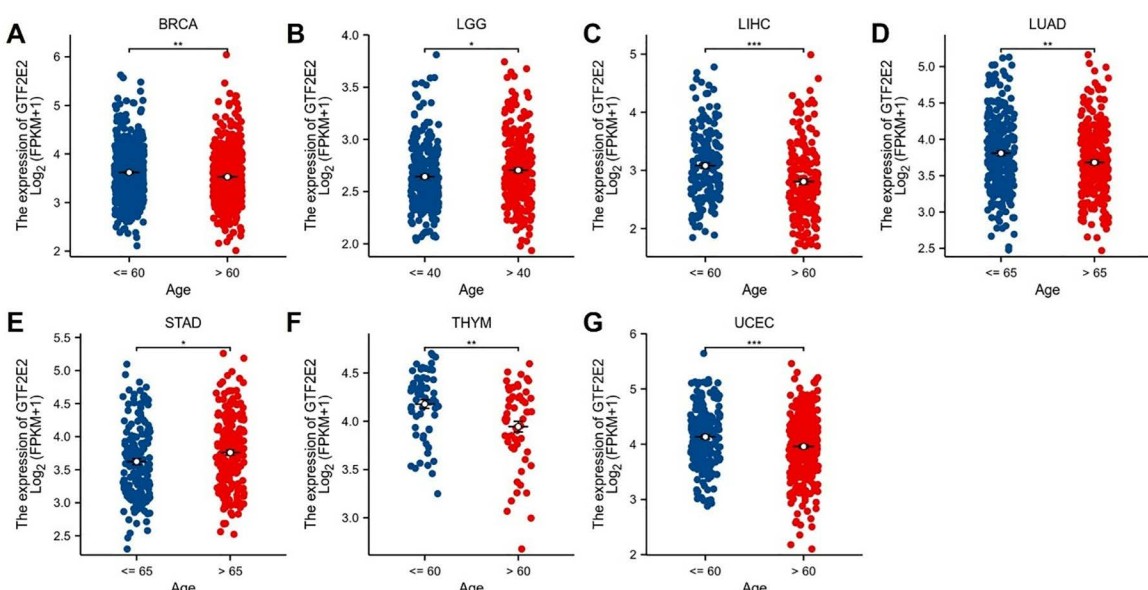

**Fig. 10. Association between GTF2E2 expression and age in (A) BRCA, (B) LGG, (C) LIHC, (D) LUAD, (E) STAD, (F) THYM and (G) UCEC.** $^*P < 0.05$, $^{**}P < 0.01$, $^{***}P < 0.001$.

## Immunogenomic analyses of GTF2E2 in the 33 cancers

To explore the relationship between the gene GTF2E2 and immune responses in cancer, we analyzed its association with various immune cells and regulatory factors across 33 different cancer types. Our analysis, depicted in a comprehensive heatmap, shows that GTF2E2 is positively correlated with several activated immune cells, including Activated CD8 T cell (Act CD8) and Activated CD4 T cell (Act CD4), Central memory CD4 T cell (Tcm CD4), Gamma delta T cell (Tgd), different types of natural killer cells, activated dendritic cells, and monocytes. However, it is generally negatively correlated with other immune cell types (Fig 15A). In specific cancer types, such as BRCA, KIRC, LGG, SARC, SKCM, and THCA, GTF2E2 is positively linked with immunostimulators, indicating a potential role in enhancing immune responses (Fig 15B). Additionally, the gene correlates positively with immunoinhibitors in CESC, LIHC, and other listed cancers, suggesting a complex role in immune regulation (Fig 15C). Regarding MHCs, we observed a positive correlation between GTF2E2 and most MHCs in BRCA, GBM, LGG, SARC, TGCT, and THCA, while a negative correlation was evident in LUAD, LUSC, and MESO (Fig 15D). Similarly, cytokine relationships are predominantly positive in cancers such as KIRC and SARC, but negative in CHOL and other types (Fig 15E). Finally, cytokine receptors mostly show negative correlations with GTF2E2 in several gastrointestinal and lung cancers, with positive associations noted in a few others like KIRC (Fig 15F). These findings underscore the multifaceted role of GTF2E2 in the immune landscape of cancer, suggesting it could be a significant factor in the immune response and potentially a target for therapeutic intervention.

## Interacting chemicals and genes of GTF2E2

In the analysis conducted using the CTD database, GTF2E2 was found to be associated with 41 distinct chemicals. Among these, 22 chemicals have been shown to increase the levels of GTF2E2, while 11 appear to decrease its expression. The impact of the remaining 8 chemicals

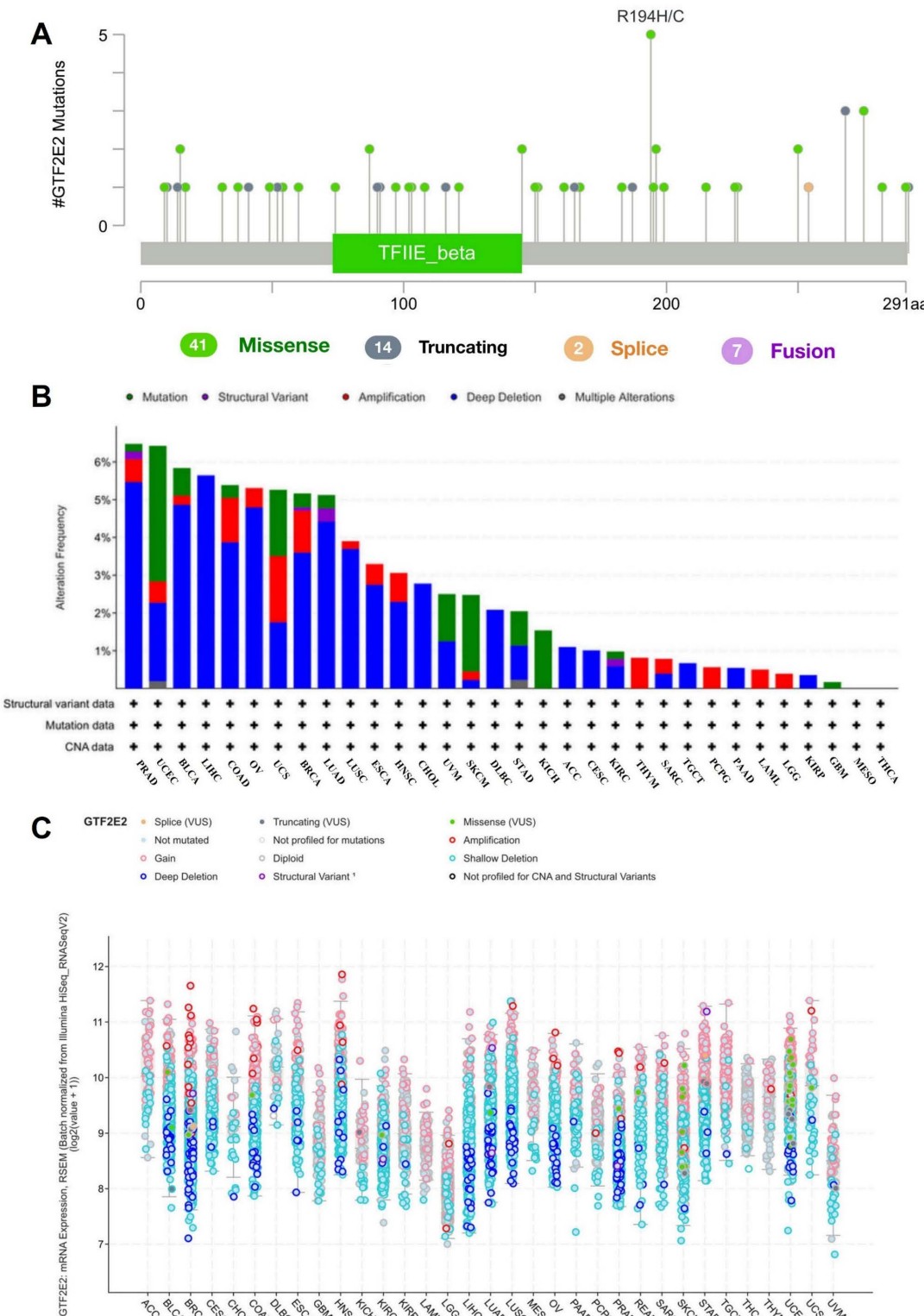

**Fig. 11. Genetic alteration of GTF2E2 across cancers.** (A) Mutation diagram of GTF2E2 across protein domains; (B) Bar chart of GTF2E2 mutations across cancers studies based on TCGA PanCancer Atlas Studies; (C) Mutation counts and types of GTF2E2 across cancers.

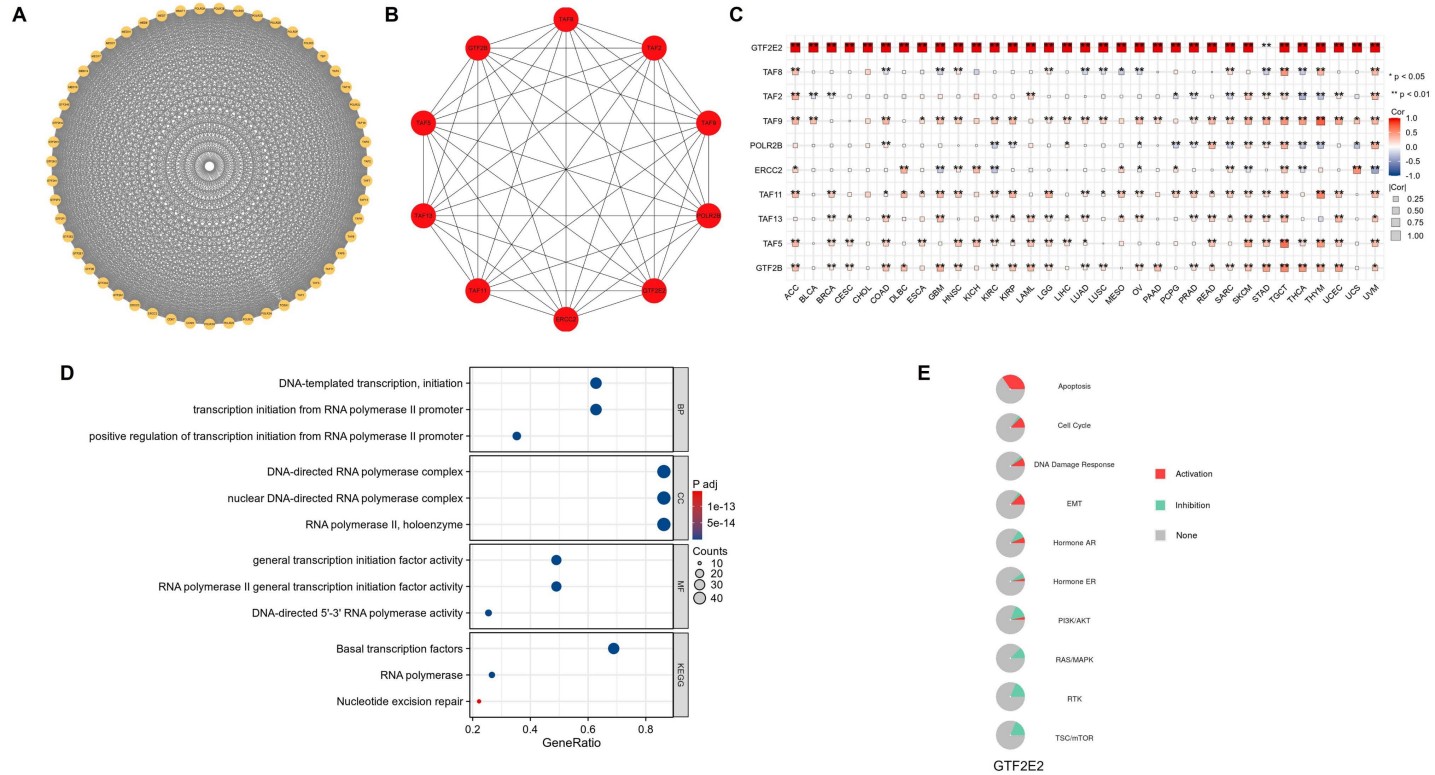

**Fig. 12. The PPI network and functional enrichment analysis of GTF2E2.** (A) The PPI network of GTF2E2, (B) The top 10 hub genes of PPI network, (C) The association hub gene with GTF2E2 in 33 cancers present as heatmap. (D) GO/KEGG pathway enrichment for GTF2E2 and closed interact genes, (E) GTF2E2 with pathway activity or inhibition. $^*P < 0.05$, $^{**}P < 0.01$.

on GTF2E2 expression remains unclear (Table 2). Furthermore, our study identified the top 20 gene relationships linked to GTF2E2 through these chemical interactions. Notably, GTF2E2 exhibits strong associations with several key genes, including Desumoylating iso-peptidase 1 (DESI1), OTU deubiquitinase 6B (OTUD6B), NGG1 interacting factor 3 like 1 (NIF3L1), sorting nexin 2 (SNX2), and NMD3 ribosome export adaptor (NMD3) (Table 3).

## Knock down of GTF2E2 induced ferroptosis and suppressed UCEC progression

In the preceding section, we conducted a comprehensive analysis of GTF2E2's expression level, diagnostic value, prognostic significance, and mutation characteristics across various cancer types. Our findings revealed that GTF2E2 was significantly upregulated in UCEC and may serve as a potential diagnostic and prognostic biomarker for this cancer type. To validate these bioinformatic results, we performed in vitro experiments focusing on UCEC. To elucidate the role of GTF2E2 in UCEC further, we knocked down GTF2E2 expression in ISK and HEC-1-A cell lines and verified the knockdown efficiency by Western blot analysis (Fig 16A). As demonstrated in (Fig 16B), the proliferative capacity of both ISK and HEC-1-A cells was progressively impaired following GTF2E2 knockdown. To assess the impact of GTF2E2 on UCEC cell migration, we conducted Transwell assays. The results revealed that GTF2E2 knockdown significantly inhibited both the migration and invasion abilities of ISK and HEC-1-A cells (Fig 16C and 16D).

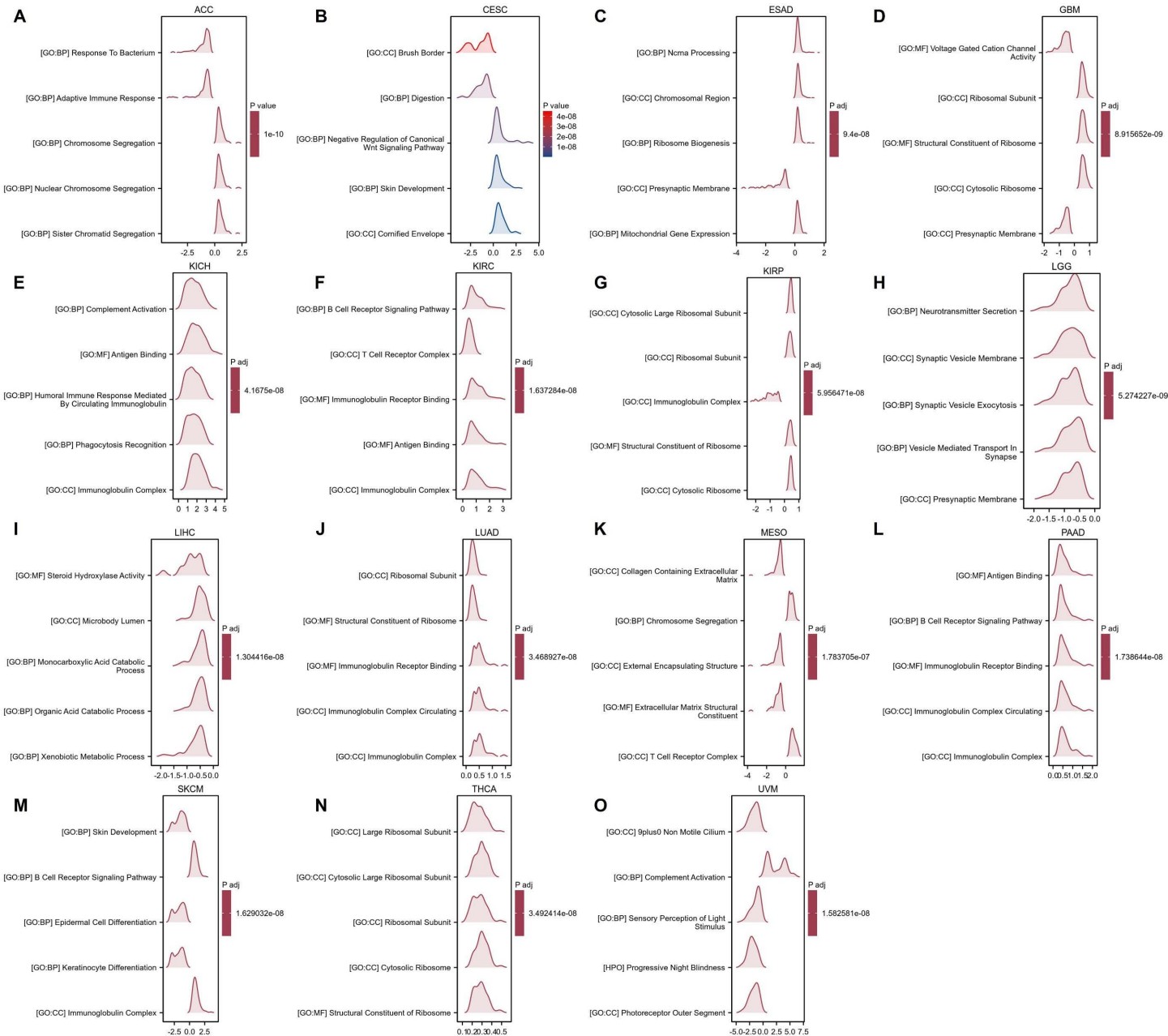

**Fig. 13. GSEA functional enrichment analysis of GTF2E2 expression in 15 cancers.** The top 5 GSEA functional enrichment pathways of GTF2E2 in (A) ACC, (B) CESC, (C) ESAD, (D) GBM, (E) KICH, (F) KIRC, (G) KIRP, (H) LGG, (I) LIHC, (J) LUAD, (K) MESO, (L) PAAD, (M) SKCM, (N) THCA, (O) UVM. The Y-axis represents one gene set and the X-axis is the distribution of logFC corresponding to the core molecules in each gene set.

To investigate the potential involvement of GTF2E2 in ferroptosis in UCEC cells, we measured the levels of ROS, LPOs, and $Fe^{2+}$ in both the GTF2E2 siRNA-treated groups (ISK-siRNA and HEC-1-A-siRNA) and their respective control groups. Our findings showed that the levels of ROS and LPOs were significantly elevated in both ISK-siRNA and HEC-1-A-siRNA groups compared to their controls (Fig 17A–17D), suggesting an enhanced oxidative stress response associated with ferroptosis following GTF2E2 knockdown. Moreover, $Fe^{2+}$ content was also markedly increased in both GTF2E2-knockdown cell groups relative to their

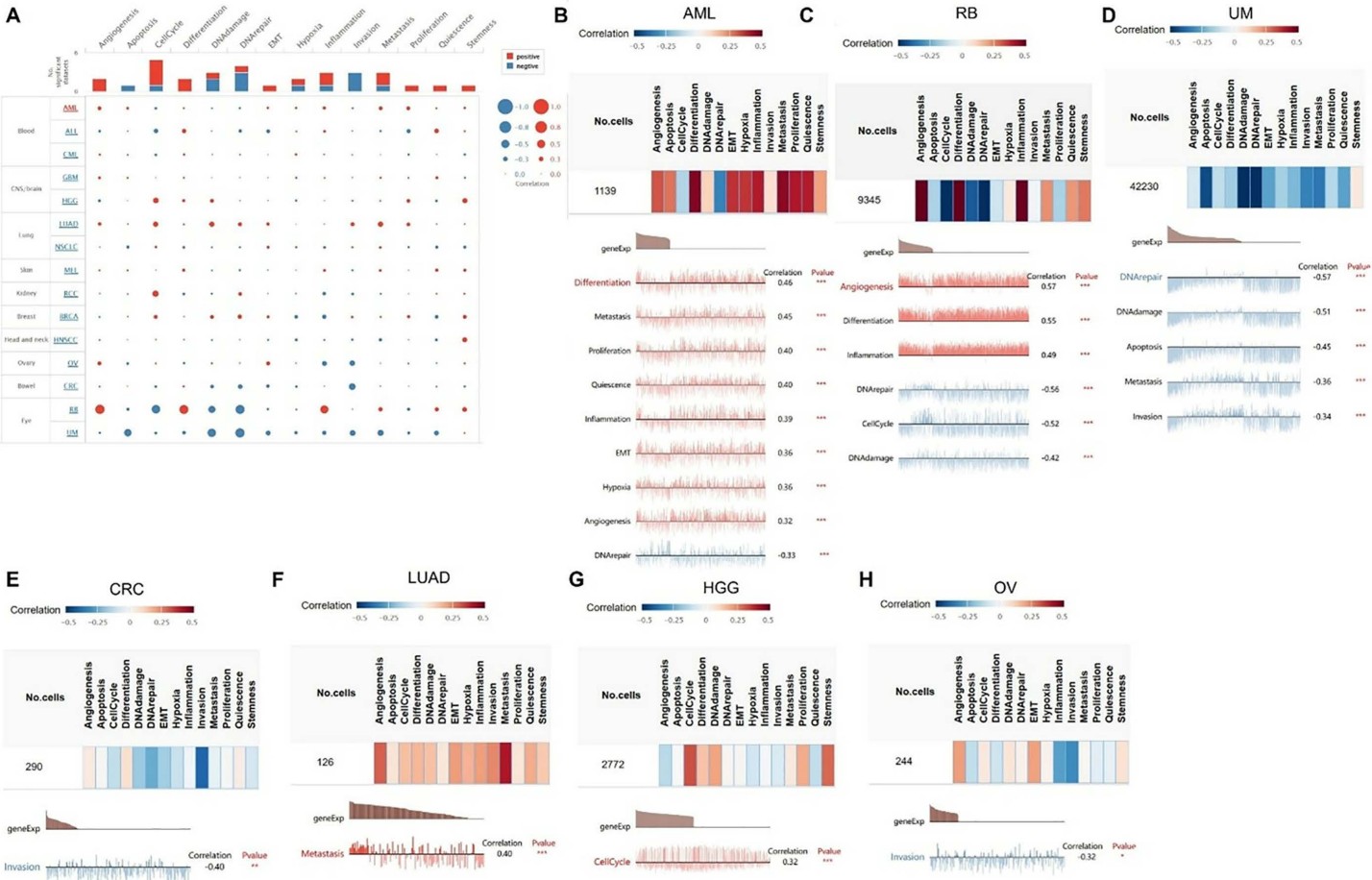

**Fig. 14. The correlation of GTF2E2 with functional state in cancers.** (A) The interactive bubble chart present correlation of GTF2E2 with functional state in 15 cancers. The correlation of GTF2E2 with functional state in (B) Acute myeloid leukemia (AML), (C) Retinoblastoma (RB), (D) Uveal Melanoma (UM), (E) Colon and Rectal Cancer (CRC), (F) LUAD, (**G**) high-grade glioma (HGG), (H) OV. X-axis represents different gene sets; *P < 0.05, **P < 0.01, ***P < 0.001.

controls (Fig 17E–17F). Finally, Western blot analysis revealed that the ISK-siRNA and HEC-1-A-siRNA groups exhibited significantly lower expression of GPX4 and higher expression of ACSL4 compared to their respective control groups (Fig 18A and 18B). We further compared the MDA levels in ISK and HEC-1-A cells following GTF2E2 knockdown. The results showed that knockdown of GTF2E2 significantly increased MDA levels (Fig 18C–18D). These findings further corroborate that GTF2E2 knockdown induced ferroptosis and inhibited UCEC progression.

## Discussion

General transcription factors come together to form a preinitiation complex, essential for proper RNA polymerase II loading at the start of transcription. A key player in this assembly is GTF2E2, a subunit of the transcription factor IIE, vital for initiating transcription and opening promoters by aiding the loading and stabilization of TFIIH [30]. GTF2E2 is involved in a range of biological functions, and changes in its expression are linked to various diseases, including trichothiodystrophy [31–33], the regulation of stem cell differentiation in spermatogonia [34] and viral replication [35]. Previous research has shown abnormal GTF2E2

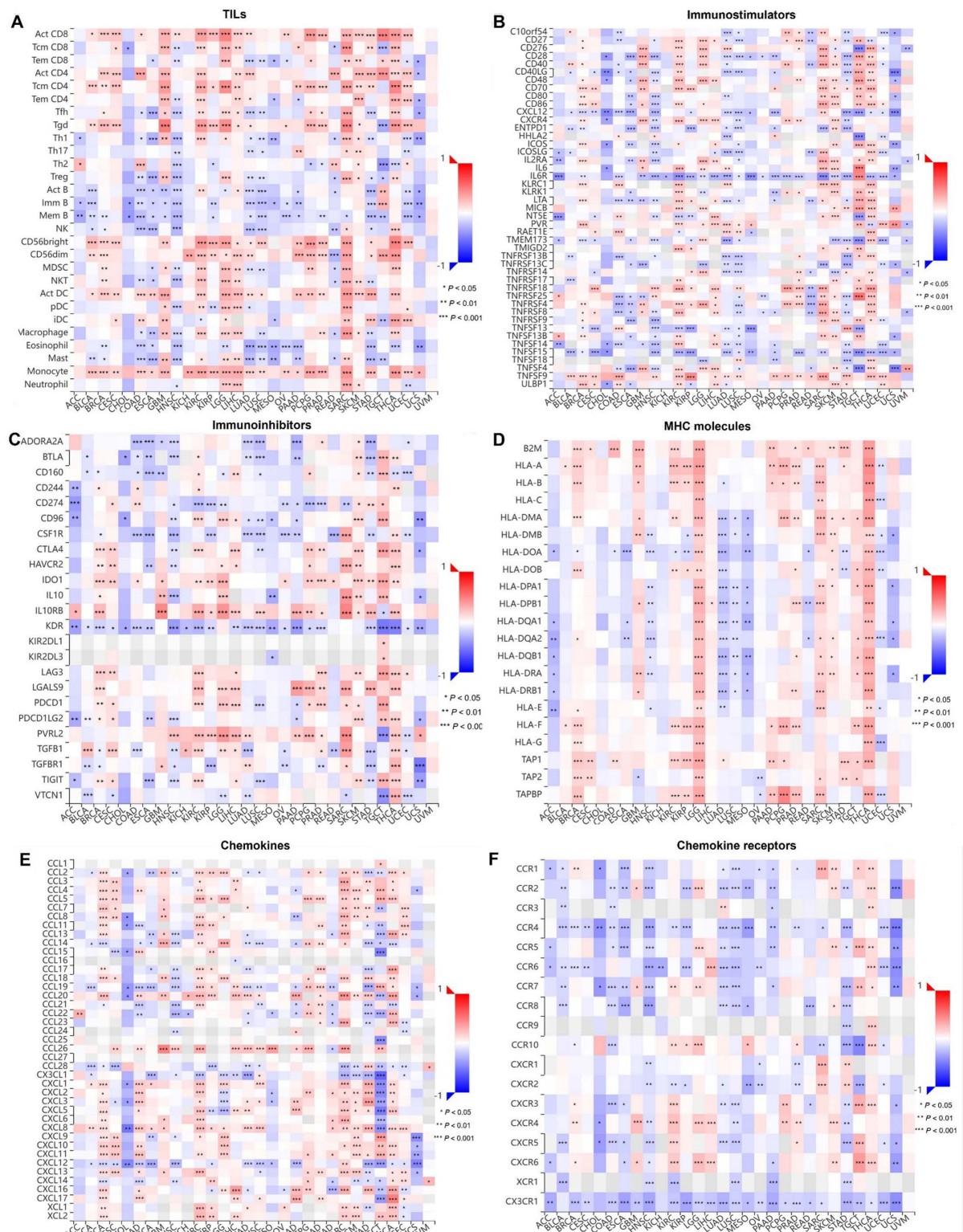

**Fig. 15. Correlation of GTF2E2 with TILs and immunoregulation-related genes in 33 cancers.** Correlations between GTF2E2 expression and (A) TILs, (B) Immunostimulators, (C) Immunoinhibitors, (D) MHC molecules, (E) Chemokines, (F) Chemokine receptors. *P < 0.05, **P < 0.01, ***P < 0.001.

**Table 2. Interacting chemicals of GTF2E2 from CTD.**

| Chemical Name | Chemical ID | Interaction Actions |
|---|---|---|
| 1,2-dimethylhydrazine | D019813 | Decreases expression |
| 2,3,7,8-tetrachlorodibenzofuran | C014211 | Increases expression |
| 2,4-dinitrotoluene | C016403 | Affects expression |
| 9,10-dimethyl-1,2-benzanthracene | D015127 | Decreases expression |
| Acetaminophen | D000082 | Affects expression |
| Aflatoxin b1 | D016604 | Increases expression |
| Benzo(a)pyrene | D001564 | Increases expression |
| Beta-methylcholine | C044887 | Affects expression |
| Bisphenol a | C006780 | Affects expression |
| Bisphenol a | C006780 | Increases expression |
| Carbon tetrachloride | D002251 | Increases expression |
| Chlordecone | D007631 | Increases expression |
| Chlorodiphenyl (54% chlorine) | D020111 | Decreases expression |
| Chlorpyrifos | D004390 | Decreases expression |
| Cisplatin | D002945 | Affects expression |
| Cyclosporine | D016572 | Increases expression |
| Decamethrin | C017180 | Decreases expression |
| Decitabine | D000077209 | Affects expression |
| Deguelin | C107676 | Increases expression |
| Dicrotophos | C000944 | Decreases expression |
| Ellagic acid | D004610 | Increases expression |
| Ethanol | D000431 | Increases expression |
| Ethinyl estradiol | D004997 | Increases expression |
| Ethyl methanesulfonate | D005020 | Decreases expression |
| Flutamide | D005485 | Increases expression |
| Formaldehyde | D005557 | Decreases expression |
| Icg 001 | C492448 | Increases expression |
| Ivermectin | D007559 | Decreases expression |
| Methidathion | C005828 | Decreases expression |
| Methyl methanesulfonate | D008741 | Increases expression |
| Nanotubes, carbon | D037742 | Increases expression |
| Phenobarbital | D010634 | Affects expression |
| Picoxystrobin | C556557 | Increases expression |
| Quercetin | D011794 | Decreases expression |
| Resveratrol | D000077185 | Increases expression |
| Rotenone | D012402 | Increases expression |
| Sodium arsenite | C017947 | Increases expression |
| Tetrachlorodibenzodioxin | D013749 | Affects expression |
| Tobacco smoke pollution | D014028 | Increases expression |
| Triptonide | C084079 | Increases expression |
| Valproic acid | D014635 | Increases expression |

expression in several types of cancer, often associated with poor outcomes. Our study uses bioinformatics to explore the role of GTF2E2 across different cancers. Our study used bioinformatics and experimental validation to explore the role of GTF2E2 in different cancers.

GTF2E2 expression is generally low in normal tissues, yet our analysis using TCGA and GTEx datasets reveals significantly higher levels in most cancer types compared to normal

**Table 3. Relationship of GTF2E2 with genes via chemical interaction, based on the CTD database.**

| Gene | Similarity Index | Common interacting chemicals |
|---|---|---|
| DESI1 | 0.3553 | 27 |
| OTUD6B | 0.3521 | 25 |
| NIF3L1 | 0.3488 | 30 |
| SNX2 | 0.3415 | 28 |
| NMD3 | 0.3372 | 29 |
| ACTR6 | 0.3333 | 27 |
| MED6 | 0.3333 | 27 |
| NUDT3 | 0.3333 | 27 |
| TM7SF3 | 0.3333 | 29 |
| UBE2M | 0.3333 | 29 |
| NUDCD2 | 0.3291 | 26 |
| ILDR2 | 0.3256 | 28 |
| POLR1E | 0.3256 | 28 |
| RPE | 0.3243 | 24 |
| GPN2 | 0.3239 | 23 |
| UBE2K | 0.3232 | 32 |
| CAB39 | 0.3226 | 30 |
| MED27 | 0.3210 | 26 |
| MINDY3 | 0.3210 | 26 |
| SNX11 | 0.3200 | 24 |

tissues. Notably, the ROC AUC values exceeded 0.6 in 24 cancer types and 0.7 in 20 types, suggesting a strong link between increased GTF2E2 expression and cancer development. Specifically, GTF2E2 shows a notable diagnostic potential with an AUC of 0.966 in GBM. Supporting this, research by Qiao et al.[17] identified GTF2E2 overexpression in glioma, correlating with poorer patient outcomes. Similarly, Zhang et al.[36] found that GTF2E2 upregulates CDC20 expression, influencing GBM severity and prognosis. These findings underscore the potential of GTF2E2 as a marker for cancer detection and progression, particularly in GBM.

To further understand GTF2E2's prognostic value, we conducted survival analyses across various cancers using Kaplan-Meier survival curves, focusing on OS, DSS, and PFI. High GTF2E2 expression is associated with better outcomes in CESC, SKCM, and THCA, but correlates with worse prognosis in several cancers, including ACC and PAAD, among others. Previous studies have associated GTF2E2 with poor prognosis in various cancers; for example, knockdown of GTF2E2 inhibits the growth and progression of lung adenocarcinoma via RPS4X in vitro and in vivo [16]. Likewise, Zhang et al. [36] demonstrated that GTF2E2 is a novel biomarker for recurrence after surgery and promotes progression of esophageal squamous cell carcinoma via miR-139-5p/GTF2E2/FUS axis, which aligns with our research findings. GTF2E2 is oncogenic in some tumours and protective in others. This may be due to the fact that GTF2E2, as a key component of the RNA polymerase II transcription initiation complex, is involved in regulating the expression of numerous downstream target genes. These target genes may include oncogenes and oncogenes, and their altered expression levels may lead to different roles of GTF2E2 in different cancer types.

The interaction between tumors and the immune system plays a crucial role in the development, progression, and treatment of cancer [37]. GTF2E2 expression varies across different cancer types, which include both molecular and immune-related subtypes. It is important to note that variations in GTF2E2 expression are observed even in cancers that

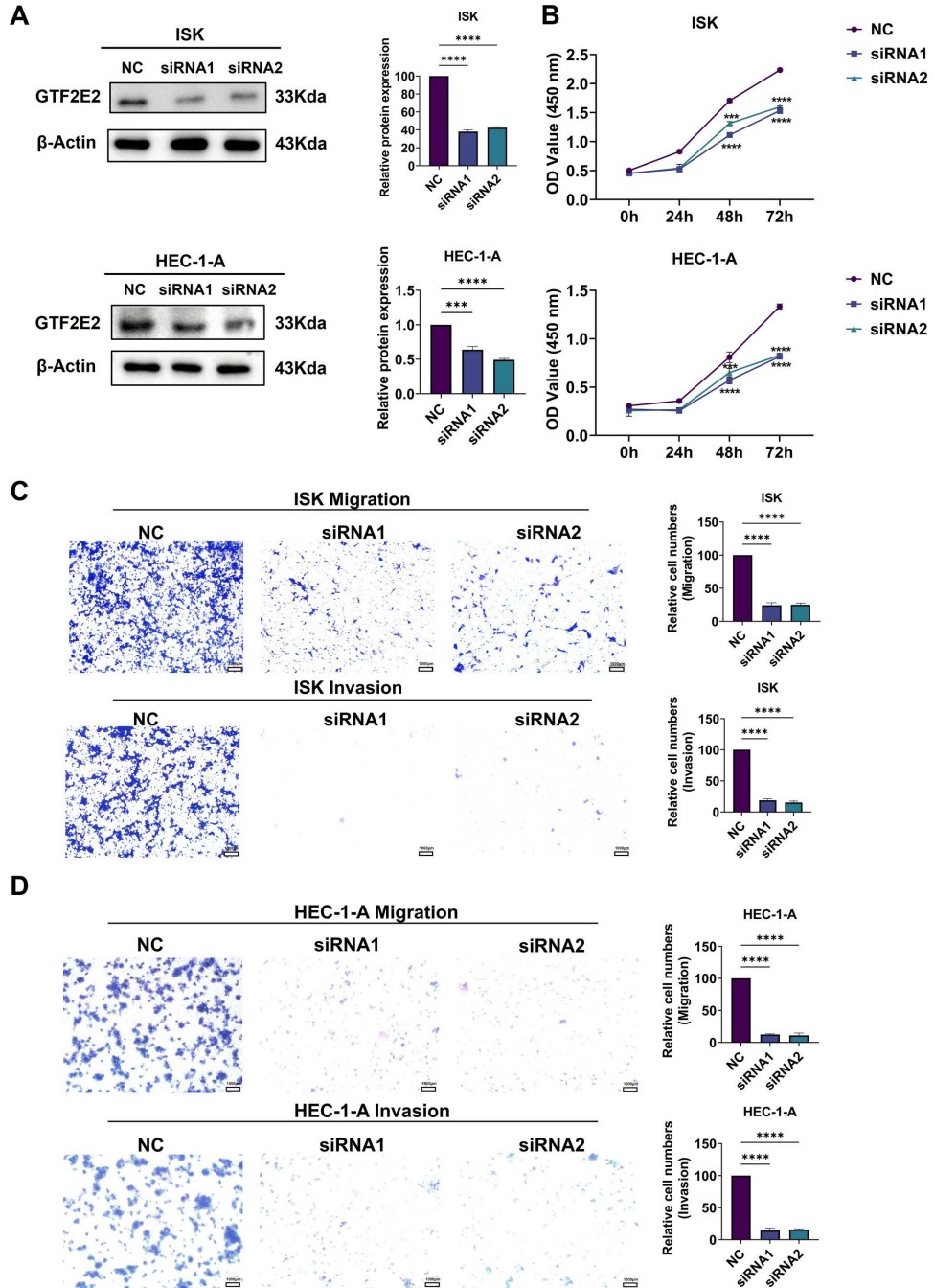

**Fig. 16. Effect of knockdown of GTF2E2 on proliferation, migration and invasion of UCEC cells.** (**A**) Knockdown of GTF2E2 protein expression level in ISK and HEC-1-A cells. (**B**) Cell viability assay of ISK and HEC-1-A cells after knockdown of GTF2E2. (**C**) Transwell assay to detect ISK, HEC-1-A migratory and invasive ability after knockdown of GTF2E2 (scale bar = 1000μm), *$P<0.05$, **$P<0.01$, ***$P<0.001$, ****$P<0.0001$.

do not primarily rely on it for predicting survival outcomes. In certain cancer subtypes, abnormal levels of GTF2E2 may lead to results that are not reflective of the overall patient population within that subtype. This inconsistency might explain why changes in GTF2E2 expression do not always correlate with survival rates in some cancers. In contrast, its

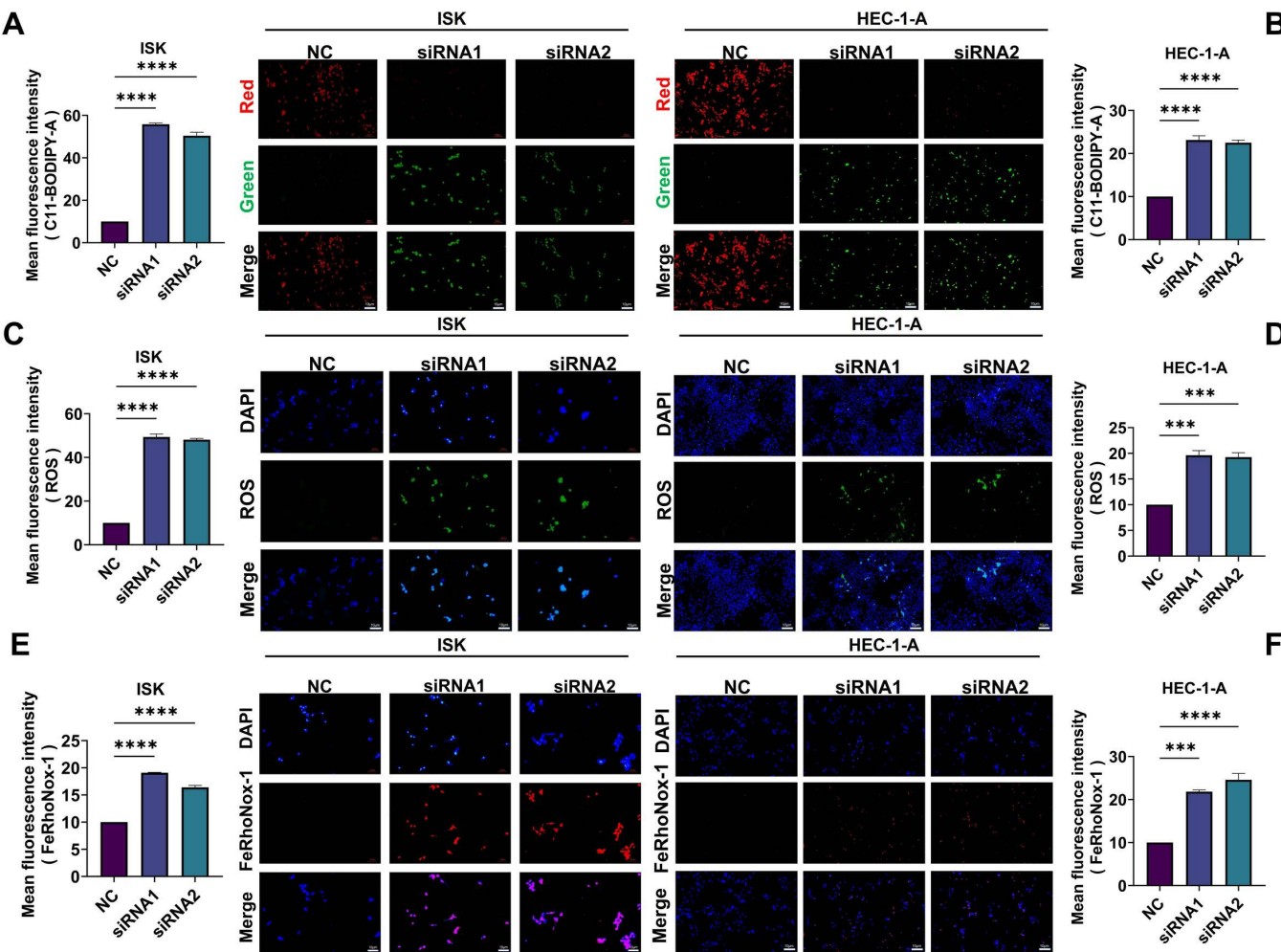

**Fig. 17. Effect of knockdown of GTF2E2 on ROS, LPOs and $Fe^{2+}$ in UCEC cells.** (**A**) ISK-siNC and/or ISK-siRNA treated cells were incubated with C11 BODIPY and detected by fluorescence microscopy. (**B**) HEC-1-A-siNC and/or HEC-1-A-siRNA treated cells were incubated with C11 BODIPY and detected by fluorescence microscopy. Fluorescence intensity of oxidised levels was detected in the 488 nm channel (green). Non-oxidised levels of fluorescence intensity were detected in the 594 nm channel (red). Scale bar = 20 μm (**C**) Effect of ISK-siNC and/or ISK-siRNA on changes in intracellular ROS levels. Scale bar = 20 μm. (**D**) Effect of HEC-1-A-siNC and/or HEC-1-A-siRNA on changes in intracellular ROS levels. Scale bar = 20 μm. (**E**) Effect of ISK-siNC and/or ISK-siRNA on changes in intracellular $Fe^{2+}$ levels. Scale bar = 20 μm. (**F**) Effect of HEC-1-A-siNC and/or HEC-1-A-siRNA on changes in intracellular $Fe^{2+}$ levels. Scale bar = 20 μm. *$P<0.05$, **$P<0.01$, ***$P<0.001$, ****$P<0.0001$.

specific levels in different molecular or immune subtypes could play a significant role in forecasting the disease's course in other cancers. Future research should thoroughly explore GTF2E2 expression across a broad range of cancer subtypes, focusing on specific molecular and immune categories.

Furthermore, we identified age-related associations with GTF2E2 expression in specific cancer types. In LGG and STAD, GTF2E2 expression was lower in younger patients, while in BRCA, LIHC, LUAD, THYM, and UCEC, higher GTF2E2 expression was linked to a younger age group. These findings may hold significance for tailoring immunotherapy strategies based on age groups [38]. Additionally, the significance of early cancer detection cannot be overstated, as it is critical for improving early diagnosis outcomes [39]. To this end, we assessed GTF2E2 expression across different cancer stages. Our findings reveal a consistent increase in GTF2E2 expression with advancing tumor stages across multiple cancer types, suggesting

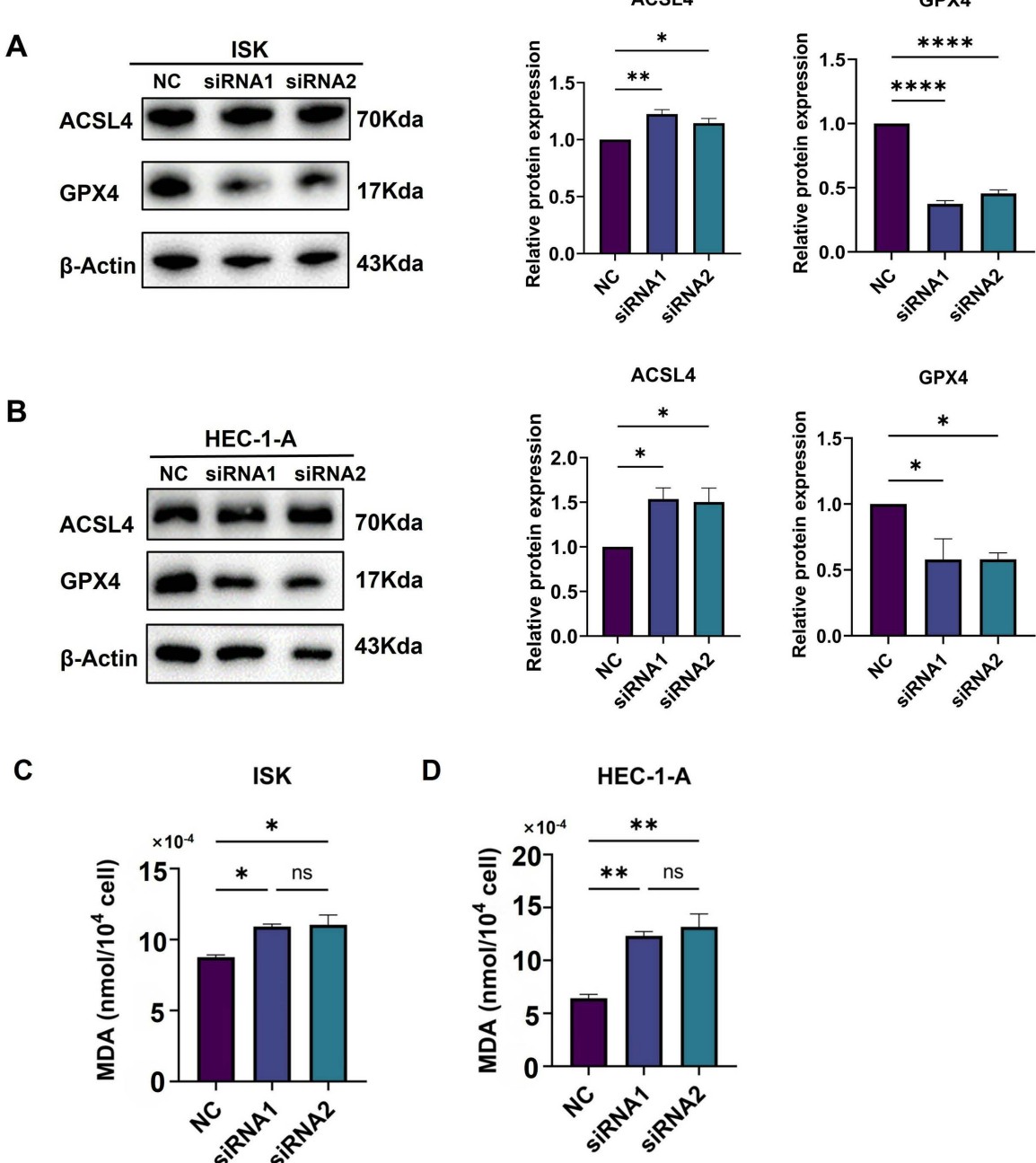

**Fig. 18. Effect of knockdown of GTF2E2 on ferroptosis-related proteins in UCEC cells.** (**A**) WB and semiquantitative analysis of ACSL4 and GPX4 after ISK-siNC and/or ISK-siRNA treatment. (**B**) WB and semiquantitative analysis of ACSL4 and GPX4 after HEC-1-A-siNC and/or HEC-1-A-siRNA treatment. (**C**) MDA levels in ISK cells following GTF2E2 knockdown. (**D**) MDA levels in HEC-1-A cells following GTF2E2 knockdown. *$P<0.05$, **$P<0.01$, ***$P<0.001$, ****$P<0.0001$.

its potential as a prognostic biomarker. This correlation underscores the value of GTF2E2 in prognosis and highlights its utility in cancer staging.

In our study, we analyzed GTF2E2 mutations across 32 cancers using the cBioPortal platform, finding low mutation frequencies overall. However, the mutations were most prevalent in prostate, uterine, and bladder cancers (PRAD, UCEC, and BLCA, respectively),

highlighting a potential link between GTF2E2 mutations and tumors of the urinogenital system. To further explore the biological functions of GTF2E2, we constructed a PPI network, identifying 10 central hub genes. Subsequent analysis showed a significant correlation between the expression of GTF2E2 and these hub genes across various cancers, suggesting their interconnected roles in oncological processes.

The GSEA enrichment analysis revealed a significant link between GTF2E2 and key immune-related pathways, including the T cell receptor complex, B cell receptor signaling pathway, and immunoglobulin complex. This suggests that GTF2E2 may play a role in regulating immune responses. Specifically, its association with the T cell receptor complex, crucial for antigen recognition by T cells, indicates a possible role in T cell activation and adaptive immunity. This is supported by existing research highlighting the role of transcription factors in guiding immune responses [40,41]. Similarly, GTF2E2's involvement in the B cell receptor signaling pathway points to its potential impact on B cell activation and the production of antibodies, essential for long-term immunity [42,43]. The link with the immunoglobulin complex, fundamental to antibody structure, further suggests GTF2E2's role in antibody-mediated immune responses.

Immune checkpoint molecules typically act to balance activation signals from costimulatory molecules, helping to maintain self-tolerance and prevent autoimmunity. Tumor cells can exploit this mechanism to inhibit T lymphocyte function, leading to T-cell dysfunction and enabling the tumor to evade immune detection [44,45]. Furthermore, the presence of immune cells within the tumor microenvironment (TME) is crucial, as it significantly impacts the clinical outcomes for cancer patients [46]. These immune cells can support tumor growth, playing a pivotal role in tumorigenesis [47]. In exploring the role of GTF2E2, a gene associated with immune regulation, we found that its expression correlates positively with the presence of $CD4^+$ and $CD8^+$ T cells across various cancers. This suggests that GTF2E2 could function as a novel immune checkpoint. Notably, research published in Nature has shown that activated $CD8^+$ T cells can increase IFN-γ production, which in turn triggers lipid peroxidation and ferroptosis in tumor cells [48]. Moreover, the interaction between cancer cells and immune cells within the TME not only facilitates tumor growth but also correlates strongly with increased GTF2E2 expression [49]. These insights suggest a mechanism by which GTF2E2 may contribute to both tumor initiation and progression, emphasizing its potential as a target in cancer therapy.

To validate these bioinformatic findings, we conducted in vitro experiments in UCEC cells. Knockdown of GTF2E2 significantly suppressed the proliferation, migration, and invasion of UCEC cells, confirming its oncogenic role in UCEC progression and validating our bioinformatic predictions. Interestingly, our experiments also revealed that GTF2E2 knockdown induced ferroptosis, a novel form of programmed cell death, in UCEC cells. Ferroptosis is characterized by an accumulation of lipid peroxidation products and an imbalance in iron metabolism [50]. We observed significantly elevated levels of ROS, LPOs, and $Fe^{2+}$ in GTF2E2-knockdown UCEC cells compared to controls. These findings suggest that GTF2E2 may play a role in regulating the cellular redox balance and iron homeostasis, which are critical determinants of ferroptosis sensitivity. Furthermore, GTF2E2 knockdown led to decreased expression of GPX4, a key antioxidant enzyme that protects cells against ferroptosis, and increased expression of ACSL4, an enzyme involved in the biosynthesis of long-chain polyunsaturated fatty acids, which are susceptible to lipid peroxidation [51]. The modulation of these ferroptosis-related proteins following GTF2E2 knockdown provides mechanistic insights into how GTF2E2 may regulate ferroptosis in UCEC cells.

This study provides a comprehensive examination of GTF2E2's role across different cancers, exploring its involvement in key signaling pathways, mutation sites, and its association

with immune cell infiltration, which were further validated through in vitro experiments. These insights suggest that GTF2E2 may act as an independent prognostic marker in various cancers, with its expression levels predicting different clinical outcomes across tumor types. Despite the comprehensive nature of our study, which combines extensive bioinformatic analyses and in vitro experimental validation, there are several limitations that should be acknowledged. First, our in vitro experiments were limited to two UCEC cell lines, which may not fully represent the heterogeneity of UCEC tumors. Future studies should include additional cell lines and primary tumor samples. Second, although we demonstrated that GTF2E2 knockdown induces ferroptosis in UCEC cells, the precise mechanisms underlying this regulation remain unclear; moreover, our current experiments did not directly quantify ferric iron or intracellular GSH levels to further elucidate this process. Further investigations into the interactions between GTF2E2 and ferroptosis-related genes and pathways are needed. Lastly, our study lacked in vivo animal models and clinical samples to validate our bioinformatic and in vitro findings. Future research should employ xenograft or genetically engineered mouse models and investigate GTF2E2 expression and prognostic significance in clinical tumor samples.

## Conclusion

In summary, we utilized various methods to demonstrate the importance of GTF2E2 across multiple cancer types. Knockdown GTF2E2 can affect the expression of GPX4 and ACSL4 to regulate ferroptosis and at the same time increase the content of LPOs, $Fe^{2+}$ and ROS levels, thereby activating the regulation of ferroptosis pathway in the development of UCEC.

## Supporting information

**S1 Fig. Receiver Operator Characteristic (ROC) curve of GTF2E2 in 4 Cancers.** Cancers with AUC < 0.7 for GTF2E2: (A) ACC, (B) PAAD, (C) PRAD, (D) STAD.
(TIF)

**S2 Fig. Correlations between GTF2E2 and DSS in 14 cancers. DSS K-M curve for GTF2E2 14 cancer types. The unit of X-axis is month.** (A) ACC, (B) CESC, (C) ESAD, (D) GBM, (E) KICH, (F) KIRC, (G) KIRP, (H) LGG, (I) LUAD, (J) LUSC, (K) MESO, (L) OSCC, (M) SKCM, (N) UVM.
(TIF)

**S3 Fig. Correlations between GTF2E2 and PFI in 12 cancers. PFI K-M curve for GTF2E2 12 cancer types. The unit of X-axis is month.** (A) ACC, (B) CESC, (C) COAD, (D) ESCC, (E) KICH, (F) KIRC, (G) KIRP, (H) LGG, (I) LIHC, (J) MESO, (K) PAAD, (L) UVM.
(TIF)

**S4 Fig. Correlations between GTF2E2 expression and immune subtype in 10 cancers.** (A) BRCA, (B) COAD, (C) ESCA, (D) LUSC, (E) PRAD, (F) READ, (G) SARC, (H) STAD, (I) TGCT, (J) UCEC.
(TIF)

## Author contributions

**Conceptualization:** Nie Zhang, Xuejin Qin, Ke Han, Di Zhang.

**Data curation:** Nie Zhang.

**Formal analysis:** Nie Zhang, Xuejin Qin, Ke Han, Di Zhang.

**Investigation:** Jingjing Liu, Zhengchun Zhu, Di Zhang.

**Methodology:** Nie Zhang, Jingjing Liu, Manman Kang, Zhengchun Zhu.

**Project administration:** Manman Kang.

**Resources:** Nie Zhang.

**Supervision:** Fei Zhong.

**Validation:** Fei Zhong.

**Writing – original draft:** Nie Zhang.

**Writing – review & editing:** Fei Zhong.

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
