## [Decision Letter · Decision Letter 0]

8 Jan 2025

Dear Dr. Zhong,

Thank you for submitting your manuscript to PLOS ONE. After careful consideration, we feel that it has merit but does not fully meet PLOS ONE’s publication criteria as it currently stands. Therefore, we invite you to submit a revised version of the manuscript that addresses the points raised during the review process.

We look forward to receiving your revised manuscript.

Kind regards,

Zu Ye, Ph.D.

Academic Editor

PLOS ONE

Journal requirements:  When submitting your revision, we need you to address these additional requirements. 1. Please ensure that your manuscript meets PLOS ONE's style requirements, including those for file naming. The PLOS ONE style templates can be found at  https://journals.plos.org/plosone/s/file?id=wjVg/PLOSOne_formatting_sample_main_body.pdf and  https://journals.plos.org/plosone/s/file?id=ba62/PLOSOne_formatting_sample_title_authors_affiliations.pdf.  2. PLOS ONE now requires that authors provide the original uncropped and unadjusted images underlying all blot or gel results reported in a submission’s figures or Supporting Information files. This policy and the journal’s other requirements for blot/gel reporting and figure preparation are described in detail at https://journals.plos.org/plosone/s/figures#loc-blot-and-gel-reporting-requirements and https://journals.plos.org/plosone/s/figures#loc-preparing-figures-from-image-files. When you submit your revised manuscript, please ensure that your figures adhere fully to these guidelines and provide the original underlying images for all blot or gel data reported in your submission. See the following link for instructions on providing the original image data: https://journals.plos.org/plosone/s/figures#loc-original-images-for-blots-and-gels.    In your cover letter, please note whether your blot/gel image data are in Supporting Information or posted at a public data repository, provide the repository URL if relevant, and provide specific details as to which raw blot/gel images, if any, are not available. Email us at plosone@plos.org if you have any questions. 3. Please amend either the title on the online submission form (via Edit Submission) or the title in the manuscript so that they are identical.  4. PLOS requires an ORCID iD for the corresponding author in Editorial Manager on papers submitted after December 6th, 2016. Please ensure that you have an ORCID iD and that it is validated in Editorial Manager. To do this, go to ‘Update my Information’ (in the upper left-hand corner of the main menu), and click on the Fetch/Validate link next to the ORCID field. This will take you to the ORCID site and allow you to create a new iD or authenticate a pre-existing iD in Editorial Manager.  5. Please note that PLOS ONE has specific guidelines on code sharing for submissions in which author-generated code underpins the findings in the manuscript. In these cases, we expect all author-generated code to be made available without restrictions upon publication of the work. Please review our guidelines at https://journals.plos.org/plosone/s/materials-and-software-sharing#loc-sharing-code and ensure that your code is shared in a way that follows best practice and facilitates reproducibility and reuse.  6. Thank you for stating the following financial disclosure:   [This work was supported by the Anhui University Natural Science Research Project (2023AH050675); Research Fund of Anhui Institute of Translational Medicine (2023zhyx-C98); Research Fund Project of Anhui Medical University (2021xkj077, 2022xkj213); Anhui Provincial Key Research and Development Project (2022e07020050); Science Research Project of Anhui Health Commission (AHWJ2021b097); Science Research Project of Anhui Health Commission (AHWJ2023A10076); Scientific Research Program of Fuyang Municipal Health Commission (FY2021-126).].   Please state what role the funders took in the study.  If the funders had no role, please state: ""The funders had no role in study design, data collection and analysis, decision to publish, or preparation of the manuscript.""  If this statement is not correct you must amend it as needed.  Please include this amended Role of Funder statement in your cover letter; we will change the online submission form on your behalf.  7. Please include captions for your Supporting Information files at the end of your manuscript, and update any in-text citations to match accordingly. Please see our Supporting Information guidelines for more information: http://journals.plos.org/plosone/s/supporting-information. 

Reviewers' comments:

Reviewer's Responses to Questions

**Comments to the Author**

1. Is the manuscript technically sound, and do the data support the conclusions?

Reviewer #1: Yes

Reviewer #2: Partly

Reviewer #3: Yes

2. Has the statistical analysis been performed appropriately and rigorously?

Reviewer #1: Yes

Reviewer #2: Yes

Reviewer #3: Yes

3. Have the authors made all data underlying the findings in their manuscript fully available?

Reviewer #1: Yes

Reviewer #2: Yes

Reviewer #3: Yes

4. Is the manuscript presented in an intelligible fashion and written in standard English?

Reviewer #1: Yes

Reviewer #2: Yes

Reviewer #3: Yes

Reviewer #1: 1. The Actin bands in the western blot are uneven in multiple places�it is recommended to replace them.

2. It is recommended to add MDA detection or glutathione detection for the ferroptosis phenotype to enhance the persuasiveness.

Reviewer #2: Comments to the Author

This manuscript is research paper about the overexpression of GTF2E2 across multiple cancer types, which contributed to proliferation and ferroptosis-regulation. This work could be an interesting and important topic to investigate, but it may have some problems that need to be addressed before submitting it to a journal. Some of the possible problems are:

1. In figure 3. G, K and J, different from the mRNA level, the expression of GTF2E2 is significantly higher in the normal tissues, the result should be fully discussed.

2. In figure 16. C, the figure given in siRNA1 and siRNA2 in ISK migration are the same set of cells (see the up-left of siRNA1 and up-right of siRNA2). The author should check the figure carefully.

3. The author chose ISK and HEC1A cells to carry out further experiment. But, the results of GTF2E2 in UCEC was not highlight in former part of the research and not shown in figure 6. The author should elaborate the differences between UCEC and other tumors.

4. In figure 17, the author used fluorescence intensity to quantify ROS, LPOs and Fe2+ level. However, the fluorescence intensity was detected using an microscope, which make the results lack of objectivity and unreliable. In addition, the ROS level was measured using a ROS Assay Kit, however, the manufacturer and the methods of ROS Assay Kit were not mentioned.

5. Based on the critical role of ACSL4 in AA and AdA metabolism and lipid peroxidation, ACSL4 can be used as a biomarker for ferroptosis and can promote ferroptosis. However, due to the different content and distribution of fatty acids in different tumor cells and different tissue cells, sensitivity of ACSL4 to ferroptosis varies greatly, the expression of ferroptosis related proteins such as FTH, xCT and TFR should be measured.

6. Based on the aforementioned content, the results did not directly reflect the role of GTF2E2 in ferric ion and GSH regulation, cannot fully support the conclusion that GTF2E2 activating the regulation of ferroptosis pathway in the development of UCEC.

Reviewer #3: Dear Editor,

I hope this message finds you well. I am writing to provide my review of the manuscript titled "Pan-Cancer Analysis and Validation Show GTF2E2’s Diagnostic, Prognostic, and Immunological Roles in Regulating Ferroptosis in Endometrial Cancer", which I had the pleasure of reviewing.

After a thorough evaluation, I believe that this manuscript should be accepted for publication. The authors have presented a well-structured and insightful study that addresses a significant gap in the current literature. The findings are not only robust but also offer valuable implications for both readers and future research in the field.

The manuscript contributes to our understanding , and I am confident that it will be a valuable resource for researchers, practitioners, and policymakers alike. The clarity of the writing and the rigor of the methodology further enhance its suitability for publication.

I appreciate the authors' efforts in conducting this research and their commitment to advancing knowledge in this area. I strongly recommend that the manuscript be accepted for publication.

Thank you for considering my review. Please feel free to reach out if you require any further information.

**Do you want your identity to be public for this peer review?** For information about this choice, including consent withdrawal, please see our Privacy Policy

Reviewer #1: No

Reviewer #2: No

Reviewer #3: **Yes: ** Muhammad Asmat Ullah Saleem

---

## [Author Response · Author response to Decision Letter 1]

12 Feb 2025

We sincerely thank the editor and all reviewers for their valuable feedback that we have used to improve the quality of our manuscript (Submission ID: PONE-D-24-58020). The reviewers’ comments are shown in italics below, with specific issues numbered. Our responses are provided in regular font.

Comments from Reviewer 1:

1. The Actin bands in the western blot are uneven in multiple places�it is recommended to replace them.

The author’s answer:

Thank you very much for taking the time to review our manuscript and for sharing your concern regarding the uneven Actin bands. We appreciate your careful observation. After re-examining the original data and experimental records, we confirmed that these variations reflect the genuine experimental results rather than technical artifacts. We acknowledge that the Actin signals show slight differences in intensity; however, the overall loading and normalization remain consistent across all samples. Consequently, we believe that replacing the Actin bands may not be strictly necessary for the integrity of the data. Additionally, we have included images displaying the original data and the normalization process for further clarity.

Nonetheless, we would be happy to provide further clarification, additional quantification data, or replicate experiments as needed to strengthen the reliability of our findings. We value your feedback and are committed to ensuring the quality and transparency of our results. Thank you again for your thoughtful review and constructive comments.

2. It is recommended to add MDA detection or glutathione detection for the ferroptosis phenotype to enhance the persuasiveness.

The author’s answer:

Thank you very much for your thoughtful suggestion to include MDA detection or glutathione measurements to strengthen the evidence for ferroptosis. We appreciate your insight and have accordingly performed MDA detection experiments. These results have now been incorporated into the revised manuscript to further support our findings on ferroptosis. We believe that the additional data reinforce our conclusions and address your concerns regarding the persuasiveness of our study.

Thank you again for your valuable feedback, and please let us know if there are any additional revisions or clarifications we can provide.

Comments from Reviewer 2:

1. In figure 3. G, K and J, different from the mRNA level, the expression of GTF2E2 is significantly higher in the normal tissues, the result should be fully discussed.

The author’s answer:

Thank you very much for bringing our attention to the discrepancy in Figure 3 (panels G, K, and J) regarding the expression of GTF2E2. We sincerely apologize for the confusion caused by our oversight. After carefully reviewing our data, we identified that the images were inadvertently placed incorrectly in the original figure. We have now replaced them with the correct images and verified their consistency with the corresponding mRNA-level data. We appreciate your careful examination of our manuscript and the opportunity to correct this mistake. The updated figure no longer shows higher GTF2E2 expression in normal tissues, and we believe it now accurately reflects the true expression patterns. Please let us know if you have any further questions or concerns about the revised figures or our discussion. Thank you again for your constructive feedback and understanding.

2. In figure 16. C, the figure given in siRNA1 and siRNA2 in ISK migration are the same set of cells (see the up-left of siRNA1 and up-right of siRNA2). The author should check the figure carefully.

The author’s answer:

Thank you for calling our attention to the incorrect images in Figure 16C (specifically the siRNA1 and siRNA2 panels for ISK migration). We sincerely apologize for this oversight. After a thorough review, we discovered that the images were indeed misplaced during figure assembly. We have now corrected them with the accurate representative fields, which align with the quantitative data presented.

We deeply regret any confusion or concern this mistake may have caused and truly appreciate your patience and understanding. Please rest assured that we have taken additional steps to review and verify all of our figures to ensure that such errors do not occur again. We value your feedback and remain committed to maintaining the highest standards of scientific rigor and integrity. Thank you for your understanding, and please let us know if you have any further questions or concerns.

3. The author chose ISK and HEC1A cells to carry out further experiment. But, the results of GTF2E2 in UCEC was not highlight in former part of the research and not shown in figure 6. The author should elaborate the differences between UCEC and other tumors.

The author’s answer:

Thank you for your valuable comments and for pointing out the need to elaborate on our rationale for selecting Uterine Corpus Endometrial Carcinoma (UCEC) for experimental validation. We appreciate your careful review and are happy to provide further clarification. Our decision to focus on UCEC was based on several key findings from our pan-cancer bioinformatics analysis:

①High Expression of GTF2E2 in UCEC

Our analysis revealed that GTF2E2 is significantly upregulated in UCEC at both the mRNA and protein levels compared to normal tissues (Fig. 2). This suggests a potential role for GTF2E2 in UCEC tumorigenesis.

②Strong Diagnostic Potential in UCEC

The ROC curve analysis demonstrated that GTF2E2 has a strong diagnostic value for UCEC (AUC = 0.779), indicating its potential as a biomarker for early detection.

③Prognostic Significance of GTF2E2 in UCEC

Kaplan-Meier survival and Cox regression analyses (Fig. 5) indicated that GTF2E2 expression correlates with clinical prognosis in UCEC, further supporting its biological importance in this cancer type.

Although UCEC is not shown separately in the survival analyses in Figure 6, we have clearly indicated the prognostic significance of UCEC in the main text (section 5.3) and highlighted its specificity in pan-cancer in the discussion section (section 7.3).

④Functional Role in Tumor Progression and Ferroptosis

Given that ferroptosis is emerging as a novel therapeutic target in cancer, we sought to investigate whether GTF2E2 regulates ferroptosis in UCEC. Our in vitro experiments confirmed that GTF2E2 knockdown significantly inhibited UCEC cell proliferation, migration, and invasion while promoting ferroptosis, as evidenced by increased ROS, lipid peroxidation (LPOs), and Fe²⁺ accumulation. These results validate our bioinformatics predictions and confirm a mechanistic role for GTF2E2 in UCEC progression.

⑤Frequent Genetic Alterations in UCEC

Our genetic alteration analysis (Fig. 11) showed that GTF2E2 mutations were among the most frequent in UCEC, further indicating its relevance in this cancer type.

⑥Association with Immune and Molecular Subtypes

Our findings demonstrated that GTF2E2 expression varies significantly across molecular and immune subtypes in UCEC (Figs. 7 & 8), suggesting that it may influence the tumor immune microenvironment and contribute to disease progression.

Based on these findings, we selected UCEC for experimental validation because GTF2E2 is highly expressed in UCEC and demonstrates strong diagnostic and prognostic potential. Additionally, our in vitro experiments confirmed that GTF2E2 plays a functional role in tumor progression and ferroptosis, further supporting its biological significance in this cancer type. Moreover, UCEC exhibits a high frequency of GTF2E2 mutations, reinforcing its relevance in tumor development. Furthermore, GTF2E2 expression is associated with distinct immune and molecular subtypes in UCEC, suggesting its potential involvement in tumor immunoregulation. Thank you again for your insightful comments, and we appreciate your time and consideration.

4. In figure 17, the author used fluorescence intensity to quantify ROS, LPOs and Fe2+ level. However, the fluorescence intensity was detected using an microscope, which make the results lack of objectivity and unreliable. In addition, the ROS level was measured using a ROS Assay Kit, however, the manufacturer and the methods of ROS Assay Kit were not mentioned.

The author’s answer:

Thank you very much for your constructive feedback on our manuscript. We value your concerns regarding the reliability and objectivity of using fluorescence imaging to quantify reactive oxygen species (ROS), lipid peroxidation (LPO), and Fe2+ levels, and we would like to address them comprehensively.

In order to ensure reproducibility and minimize technical variation, we standardized all microscope settings-such as excitation intensity, exposure time, and gain-across experimental groups. We then performed quantitative image analysis by selecting identical regions of interest and applying the same background subtraction for every sample. Furthermore, each experiment was conducted in triplicate (and captured from multiple fields in each replicate) to bolster the statistical robustness of our findings and confirm the consistency of the results.

We also acknowledge the need for detailed information on the assays. In our revised manuscript, we have specified the manufacturer and catalog numbers for all key reagents and antibodies, including the Reactive Oxygen Species Assay Kit (S0033S) purchased from Beyotime (Shanghai, China). To further support our fluorescence-based measurements, we performed Western blot analysis of key ferroptosis-related proteins, ACSL4 and GPX4. Changes in these protein levels corroborate the observed differences in ROS, LPO, and Fe2+ signals captured by fluorescence microscopy. Moreover, we supplemented our experiments with a malondialdehyde (MDA) assay-an established biochemical marker of lipid peroxidation-which confirmed the fluorescence results and solidified our conclusions. The alignment of these independent methods demonstrates the rigor and validity of our data.

We will include all relevant details-such as the manufacturer, catalog number, and protocol-of the kits and antibodies used throughout our study in the revised manuscript to ensure clarity and transparency for readers. Thank you once again for your valuable suggestions, and we appreciate the opportunity to refine and strengthen our work in response to your comments.

5.Based on the critical role of ACSL4 in AA and AdA metabolism and lipid peroxidation, ACSL4 can be used as a biomarker for ferroptosis and can promote ferroptosis. However, due to the different content and distribution of fatty acids in different tumor cells and different tissue cells, sensitivity of ACSL4 to ferroptosis varies greatly, the expression of ferroptosis related proteins such as FTH, xCT and TFR should be measured.

The author’s answer:

Thank you for highlighting the importance of investigating additional ferroptosis-related proteins such as FTH, xCT, and TFR. We fully agree that these markers provide valuable insights into iron homeostasis and cysteine import and could further strengthen the overall understanding of ferroptosis. However, our present study primarily focuses on the lipid metabolism arm of ferroptosis, centered around ACSL4 and GPX4. In addition to these protein markers, we performed an MDA assay to confirm lipid peroxidation, a hallmark of ferroptotic cell death. We believe these experiments collectively support our conclusions regarding ferroptosis induction in our model. Nevertheless, we appreciate your suggestion and will certainly consider incorporating a broader panel of ferroptosis-related markers in future studies to complement and extend our current findings. Thank you again for your constructive feedback, which has been invaluable in guiding our research directions.

6. Based on the aforementioned content, the results did not directly reflect the role of GTF2E2 in ferric ion and GSH regulation, cannot fully support the conclusion that GTF2E2 activating the regulation of ferroptosis pathway in the development of UCEC.

The author’s answer:

Thank you for sharing your concerns regarding the direct evidence of GTF2E2’s role in regulating ferric ion and GSH, and whether our data fully support the conclusion that GTF2E2 activates the ferroptosis pathway in UCEC. We would like to address these points based on our current findings, while also acknowledging the limitations of our study.

First, our data provide several lines of evidence suggesting that GTF2E2 is involved in ferroptosis. We observed changes in key ferroptosis markers (e.g. GPX4, ACSL4) after modulating GTF2E2 expression. In addition, we performed malondialdehyde (MDA) assays and fluorescence-based ROS, LPO, and Fe2+ measurements, which consistently indicated heightened oxidative stress and lipid peroxidation - a hallmark of ferroptosis - when GTF2E2 expression was altered. These results collectively point toward a role for GTF2E2 in sensitizing UCEC cells to ferroptotic cell death.

Nevertheless, we acknowledge that our current experiments did not directly quantify ferric iron or intracellular GSH levels in the context of GTF2E2 manipulation. We have added this point to the limitations section of the revised manuscript, where we discuss that the precise mechanisms by which GTF2E2 might affect ferric ion and GSH remain to be fully elucidated. Future work will include direct measurements of ferric iron and GSH, as well as additional functional studies, to elucidate the intricate relationship between GTF2E2 and these regulators of ferroptosis.

We appreciate your feedback, as it highlights avenues for further investigation and strengthens our commitment to clarifying the multifaceted role of GTF2E2 in UCEC development. Thank you again for helping us improve our manuscript.

Comments from Reviewer 3:

I hope this message finds you well. I am writing to provide my review of the manuscript titled "Pan-Cancer Analysis and Validation Show GTF2E2’s Diagnostic, Prognostic, and Immunological Roles in Regulating Ferroptosis in Endometrial Cancer", which I had the pleasure of reviewing.

After a thorough evaluation, I believe that this manuscript should be accepted for publication. The authors have presented a well-structured and insightful study that addresses a significant gap in the current literature. The findings are not only robust but also offer valuable implications for both readers and future research in the field.

The manuscript contributes to our understanding , and I am confident that it will be a valuable resource for researchers, practitioners, and policymakers alike. The clarity of the writing and the rigor of the methodology further enhance its suitability for publication.

I appreciate the authors' efforts in conducting this research and their commitment to advancing knowledge in this area. I strongly recommend that the manuscript be accepted for publication.

Thank you for considering my review. Please feel free to reach out if you require any further information.

The author’s answer:

We are truly honored and grateful for your kind words and for recommending our manuscript for publication. Thank you for recognizing the value of our work and for your thoughtful appraisal of its structure, methodology, and contributions. Your positive feedback encourages us to continue our research in this field and to explore further avenues for understanding and application. We greatly appreciate the time and effort you devoted to reviewing our manuscript. Your support and insight will undoubtedly inspire us as we move forward. Please feel free to let us know if there is any additional information we can provide. Once again, thank you for your endorsement and for your invaluable contribution to our manuscript’s successful publication.

---

## [Decision Letter · Decision Letter 1]

10 Mar 2025

Dear Dr. Zhong,

Thank you for submitting your manuscript to PLOS ONE. After careful consideration, we feel that it has merit but does not fully meet PLOS ONE’s publication criteria as it currently stands. Therefore, we invite you to submit a revised version of the manuscript that addresses the points raised during the review process.

**The author stated in the revision comments that the images for Figure 3 (G, K, J) had been replaced, but the submitted revised figures show no differences from the original ones. Please verify the figures.**plosone@plos.org . A rebuttal letter that responds to each point raised by the academic editor and reviewer(s). You should upload this letter as a separate file labeled 'Response to Reviewers'.A marked-up copy of your manuscript that highlights changes made to the original version. You should upload this as a separate file labeled 'Revised Manuscript with Track Changes'.An unmarked version of your revised paper without tracked changes. You should upload this as a separate file labeled 'Manuscript'.

We look forward to receiving your revised manuscript.

Kind regards,

Zu Ye, Ph.D.

Academic Editor

PLOS ONE

**Journal Requirements:**

Reviewers' comments:

Reviewer's Responses to Questions

**Comments to the Author**

Reviewer #2: (No Response)

Reviewer #3: All comments have been addressed

2. Is the manuscript technically sound, and do the data support the conclusions?

Reviewer #2: Yes

Reviewer #3: Yes

3. Has the statistical analysis been performed appropriately and rigorously?

Reviewer #2: Yes

Reviewer #3: Yes

4. Have the authors made all data underlying the findings in their manuscript fully available?

Reviewer #2: Yes

Reviewer #3: Yes

5. Is the manuscript presented in an intelligible fashion and written in standard English?

Reviewer #2: Yes

Reviewer #3: Yes

**Reviewer #2:**  The author stated in the revision comments that the images for Figure 3 (G, K, J) had been replaced, but the submitted revised figures show no differences from the original ones. Please verify the figures.

**Reviewer #3: ** Subject: Review of Manuscript: Acceptance for Publication

Dear Editor/Authors,

I am writing to inform you that I have completed my review of the manuscript titled " Pan-Cancer Analysis and Validation Show GTF2E2's Diagnostic, Prognostic, and Immunological Roles in Regulating Ferroptosis in Endometrial Cancer” and I am pleased to accept it for publication in its current form. After careful consideration, I have found the manuscript to be well-written, well-structured, and of high quality. The authors have presented their research in a clear and concise manner, and the manuscript meets all the journal's requirements.

I have read the reviewers comments, authors well addressed the comments and suggestions, and I believe the manuscript should be accepted for publication. I have checked the manuscript for its originality, relevance, and impact, and I am confident that it will make a significant contribution to the field.

Please let me know if you need any further information from me. I appreciate the opportunity to review this manuscript and look forward to seeing it in print.

Best regards,

Muhammad Asmat Ullah Saleem

**Do you want your identity to be public for this peer review?** For information about this choice, including consent withdrawal, please see our Privacy Policy

Reviewer #2: No

Reviewer #3: **Yes: ** Muhammad Asmat Ullah Saleem

---

## [Author Response · Author response to Decision Letter 2]

11 Mar 2025

Thank you very much for taking the time to review our manuscript and for providing invaluable feedback regarding the images in Figure 3 (G, K, J). We truly appreciate your thorough assessment and the opportunity to clarify our work.

We wish to explain that our immunohistochemistry (IHC) results for GTF2E2 were obtained from the publicly available Human Protein Atlas (HPA) database (https://www.proteinatlas.org/). Following your comments, we re-examined the HPA data for both normal and cancer tissues of KIRC, THCA, and STAD. Upon careful review, we found that the differences in GTF2E2 expression in these tissues were indeed not very pronounced. Realizing that retaining these images might cause confusion and potentially mislead readers regarding the clarity of GTF2E2 expression, we have decided to remove the IHC images for KIRC, THCA, and STAD.

We confirm that removing these images does not affect our overall findings or conclusions. Nonetheless, we sincerely apologize for any inconvenience or confusion this may have caused. We take full responsibility for our oversight and lack of rigor in verifying the images before resubmission. Your comments have prompted us to scrutinize our figures more carefully, and we have taken the necessary steps to ensure the accuracy and clarity of our data going forward.

Thank you again for your constructive suggestions and for giving us the opportunity to improve our manuscript. We appreciate your understanding and patience, and we remain committed to maintaining the highest standards of research and publication ethics.

---

## [Editor Report · Decision Letter 2]

17 Mar 2025

Pan-Cancer Analysis and Validation Show GTF2E2's Diagnostic, Prognostic, and Immunological Roles in Regulating Ferroptosis in Endometrial Cancer

PONE-D-24-58020R2

Dear Dr. Zhang,

We’re pleased to inform you that your manuscript has been judged scientifically suitable for publication and will be formally accepted for publication once it meets all outstanding technical requirements.

Kind regards,

Zu Ye, Ph.D.

Academic Editor

PLOS ONE

---

## [Editor Report · Acceptance letter]

PONE-D-24-58020R2

PLOS ONE

Dear Dr. Zhong,

I'm pleased to inform you that your manuscript has been deemed suitable for publication in PLOS ONE. Congratulations! Your manuscript is now being handed over to our production team.

Kind regards,

on behalf of

Prof. Zu Ye

Academic Editor

PLOS ONE